# Edge Prompt Tuning for Graph Neural Networks

**Xingbo Fu**
University of Virginia
xf3av@virginia.edu

**Yinhan He**
University of Virginia
nee7ne@virginia.edu

**Jundong Li**
University of Virginia
jundong@virginia.edu

## Abstract

Pre-training powerful Graph Neural Networks (GNNs) with unlabeled graph data in a self-supervised manner has emerged as a prominent technique in recent years. However, inevitable objective gaps often exist between pre-training and downstream tasks. To bridge this gap, graph prompt tuning techniques design and learn graph prompts by manipulating input graphs or reframing downstream tasks as pre-training tasks without fine-tuning the pre-trained GNN models. While recent graph prompt tuning methods have proven effective in adapting pre-trained GNN models for downstream tasks, they overlook the crucial role of edges in graph prompt design, which can significantly affect the quality of graph representations for downstream tasks. In this study, we propose EdgePrompt, a simple yet effective graph prompt tuning method from the perspective of edges. Unlike previous studies that design prompt vectors on node features, EdgePrompt manipulates input graphs by learning additional prompt vectors for edges and incorporates the edge prompts through message passing in the pre-trained GNN models to better embed graph structural information for downstream tasks. Our method is *compatible* with prevalent GNN architectures pre-trained under various pre-training strategies and is *universal* for different downstream tasks. We provide comprehensive theoretical analyses of our method regarding its capability of handling node classification and graph classification as downstream tasks. Extensive experiments on ten graph datasets under four pre-training strategies demonstrate the superiority of our proposed method against six baselines. Our code is available at https://github.com/xbfu/EdgePrompt.

## 1 Introduction

Recent years have witnessed the remarkable success of Graph Neural Networks (GNNs) (Kipf & Welling, 2017; Hamilton et al., 2017; Veličković et al., 2018; Xu et al., 2019; Wu et al., 2019; Chen et al., 2020; Wang et al., 2023b) for modeling ubiquitous graph-structured data in various real-world scenarios, including social networks (Wei et al., 2023; Zhou et al., 2023), point cloud analysis (Wang et al., 2019; Zhou et al., 2021), and healthcare systems (Fu et al., 2023; Wan et al., 2024a; Liu et al., 2024). Such success is mainly attributed to their impressive capability to incorporate node features and graph structures into the representations of graph data. Generally, GNN models are trained for specific downstream tasks in an end-to-end manner. Nevertheless, the end-to-end manner for training powerful GNN models usually encounters significant challenges in practical deployments (Hu et al., 2020b; Sun et al., 2022a; Liu et al., 2023; Fang et al., 2023). First, annotating a sufficient number of labels for graph data is typically time-consuming and resource-intensive in the real world. Second, well-trained GNN models cannot be well generalized to other tasks, even on the same graph data (Wang et al., 2024b). To grapple with these critical challenges, applying pre-training techniques on graph data has become increasingly prevalent.

Numerous recent studies have focused on designing effective pre-training strategies for training powerful GNN models without using any label information from downstream tasks (Veličković et al., 2019; Hu et al., 2020b; You et al., 2020; Hou et al., 2022; Xia et al., 2022; Wang et al., 2024a; Wan et al., 2024b). The philosophy behind these pre-training strategies is to first train a GNN model on pre-training tasks via self-supervised learning and subsequently transfer the pre-trained GNN model to specific downstream tasks. Generally, there exists inevitable objective gaps between

Table 1: A brief comparison of graph prompt tuning methods in the existing studies. (PT=Pre-training, DT=Downstream task)

| Method | PT Compatibility | DT Universality | Prompt Insertion |
|---|---|---|---|
| GPPT (Sun et al., 2022a) | ✗ | ✗ | Task Embedding |
| GraphPrompt (Liu et al., 2023) | ✓ | ✓ | Readout |
| GraphPrompt+ (Yu et al., 2024b) | ✓ | ✓ | Hidden Representation |
| ALL-in-one (Sun et al., 2023) | ✓ | ✓ | Node Feature |
| GPF-plus (Fang et al., 2023) | ✓ | ✓ | Node Feature |
| MultiGPrompt (Yu et al., 2024d) | ✗ | ✓ | Hidden Representation |
| **EdgePrompt+ (Ours)** | ✓ | ✓ | **Edge Aggregation** |

pre-training and the downstream tasks. For example, the GNN model can be pre-trained for link prediction via self-supervised learning, while the downstream task may be node classification. To bridge the objective gap between pre-training and downstream tasks, we typically need to adapt the pre-trained GNN model for downstream tasks by either fine-tuning or graph prompt tuning. During fine-tuning, the parameters of the pre-trained GNN model are updated for downstream tasks (Huang et al., 2024; Zhili et al., 2024; Sun et al., 2024). Unlike fine-tuning, graph prompt tuning usually keeps the pre-trained GNN model frozen and instead trains graph prompts for downstream tasks (Sun et al., 2022a; Liu et al., 2023; Fang et al., 2023; Sun et al., 2023; Tan et al., 2023; Yu et al., 2024b; Ma et al., 2024; Yu et al., 2024a; Li et al., 2025).

While recent graph prompt tuning methods show great prowess in adapting pre-trained GNN models for various downstream tasks, the existing methods still have several fundamental limitations. First, a few studies (Sun et al., 2022a; Yu et al., 2024d) design graph prompt tuning methods based on specific pre-training strategies, which hinders their application to off-the-shelf pre-trained GNN models. Second, the important dependency information carried by graph structures is ignored in the existing studies (Fang et al., 2023; Liu et al., 2023; Sun et al., 2023). As illustrated in Table 1, these methods focus on designing and learning graph prompts primarily by applying them to node features or node representations. In this scenario, graph prompts are unable to enhance pre-trained GNN models in capturing complex graph structural information for downstream tasks.

Although the significant role of edges in graph learning has been amplified by a cornucopia of studies (Schlichtkrull et al., 2018; Gong & Cheng, 2019; Vashishth et al., 2020; Yang & Li, 2020), unfortunately, none of the existing studies have exploited edges for graph prompt tuning. Naturally, we may ask a question: *how can we devise an edge-level graph prompt tuning method to effectively enhance the performance of a pre-trained GNN model for downstream tasks?* In this study, we aim to answer this question through a pioneering investigation into designing edge prompts for downstream tasks. In our investigation, we need to overcome two key challenges. First, edge prompt design needs to be *universal*, capable of handling graphs of varying sizes and different downstream tasks, such as node classification and graph classification. Second, edge prompt design must be *compatible* with prevalent GNN models pre-trained by various strategies, especially with those that cannot accommodate edge attributes. These two challenges make the edge prompt design nontrivial, requiring an ingenious approach to graph prompt tuning.

To address the above issues, we propose a novel graph prompt tuning method named EdgePrompt purely from the perspective of edges, fundamentally differing from node-level prompt designs in the existing studies (Sun et al., 2023; Fang et al., 2023). The intuition of EdgePrompt is to manipulate the input graph by adding extra learnable prompt vectors to edges and thereby enhance the capability of pre-trained GNN models for downstream tasks. In EdgePrompt, all the edges in the input graph learn a shared prompt vector at each layer of the pre-trained GNN model. The edge prompts will be aggregated along with node representations during the forward pass of the message-passing mechanism. To further enhance the capacity of edge prompts, we propose an advanced version EdgePrompt+ that enables each edge to learn its customized prompt vectors. We provide theoretical analyses to support that our proposed method has the capability of enhancing the pre-trained GNN models for downstream tasks. We conduct extensive experiments over ten graph datasets under four pre-training strategies. The results validate the superiority of our proposed method compared with six baselines. Our contributions to this study can be summarized as follows:

- We devise a simple yet effective graph prompt tuning method, EdgePrompt and its variant EdgePrompt+, from the perspective of edges to narrow the objective gap between pre-training and downstream tasks.

- We provide comprehensive analyses of our method regarding its capability of handling various downstream tasks, including node classification and graph classification.

- We conduct extensive experiments over ten datasets under four pre-training strategies to evaluate the effectiveness of our proposed method. Experimental results demonstrate the superiority of our method compared with six baselines for both node classification and graph classification tasks.

## 2 RELATED WORK

**Graph Pre-training.** Numerous studies have proposed to train powerful GNN models via self-supervised learning (Veličković et al., 2019; Sun et al., 2020; Hu et al., 2019; You et al., 2020; Jin et al., 2020; Rossi et al., 2020; Xia et al., 2022; Hou et al., 2022; Wang et al., 2023a). These studies can be roughly categorized into two genres: contrastive methods and generative methods. Contrastive methods typically aim to maximize the agreement between augmented instances of the same object. For instance, DGI (Veličković et al., 2019) and InfoGraph (Sun et al., 2020) adopt the mutual information maximization between the local augmented instances and the global representation. GraphCL (You et al., 2020) maximizes the agreement between two views of the same graph by different augmentation strategies. SimGRACE (Xia et al., 2022) uses GNN models with perturbed parameters to obtain contrastive views without data augmentation. In the meantime, generative methods attempt to pre-train GNN models by reconstructing specific information in the input graph. For example, GraphMAE (Hou et al., 2022) pre-trains GNNs by reconstructing masked node features. In addition, edge prediction is also employed as the pre-training technique by a cornucopia of studies (Rossi et al., 2020; Jin et al., 2020; Sun et al., 2022a; Liu et al., 2023).

**Graph Prompt Tuning.** To bridge the gap between pre-training and downstream tasks, graph prompt tuning methods modify the input graph with learnable prompt vectors for downstream tasks, while keeping the pre-trained GNN model frozen. For example, GPF-plus (Fang et al., 2023) transforms the input graph to a prompted one by adding extra learnable prompt vectors to node features for downstream tasks. All-in-one (Sun et al., 2023) unifies various downstream tasks as graph-level tasks and similarly learns prompt vectors that are added to node features. GPPT (Sun et al., 2022a) mainly focuses on node classification as the downstream task and adopts link prediction as the pre-training strategy. It narrows the gap between pre-training and downstream tasks by converting node classification to link prediction. GraphPrompt (Liu et al., 2023) designs graph prompts as a feature weighting vector to obtain task-specific (sub)graph-level representations. MultiGPrompt (Yu et al., 2024d) chooses to insert prompt vectors into node representations at each hidden layer. However, all the aforementioned studies ignore the role of edges when designing graph prompts, which are widely regarded as fundamental properties in graph data.

## 3 PRELIMINARIES

Let $\mathcal{G} = (\mathcal{V}, \mathcal{E})$ denote a graph where $\mathcal{V} = \{v_1, v_2, \cdots, v_N\}$ is the set of $N$ nodes, and $\mathcal{E}$ is the edge set. $\boldsymbol{X} \in \mathbb{R}^{N \times D}$ denotes the node feature matrix where the $i$-th row $\boldsymbol{x}_i$ represents a $D$-dimensional feature vector of node $v_i \in \mathcal{V}$. The edges in $\mathcal{G}$ can also be represented by an adjacency matrix $\boldsymbol{A} \in \{0, 1\}^{N \times N}$ where each entry $a_{ij} = 1$ if $(v_i, v_j) \in \mathcal{E}$, otherwise $a_{ij} = 0$. Generally, GNN models aim to learn expressive node representations through the message-passing mechanism (Kipf & Welling, 2017; Veličković et al., 2018; Hamilton et al., 2017; Xu et al., 2019) where the representation of a target node is iteratively updated by aggregating the representations of its neighboring nodes. Specifically, a GNN model has two fundamental operators: $\text{AGG}(\cdot)$ extracting the neighboring information of the node, and $\text{COMB}(\cdot)$ integrating the previous representation of the node and its neighboring information. Mathematically, the $l$-th layer of an $L$-layer GNN model $f$ updates the representation of node $v_i \in \mathcal{V}$ by

$$\boldsymbol{h}_i^{(l)} = \text{COMB}^{(l)}(\boldsymbol{h}_i^{(l-1)}, \text{AGG}^{(l)}(\{\boldsymbol{h}_j^{(l-1)} : v_j \in \mathcal{N}(v_i)\})), \tag{1}$$

(a) **Learning prompt vectors on nodes**   (b) **Learning prompt vectors on edges**

Figure 1: Learning prompt vectors on a node may uniformly pass them to its neighboring nodes while learning prompt vectors on edges can result in customized prompt aggregation.

where $\boldsymbol{h}_i^{(l)} \in \mathbb{R}^{D_l}$ denotes the $D_l$-dimensional representation of node $v_i$ at the $l$-th layer, and $\mathcal{N}(v_i)$ denotes the neighbors of node $v_i$. $\boldsymbol{h}_i^{(0)} \in \mathbb{R}^D$ is initialized with node $v_i$'s feature $\boldsymbol{x}_i$. The final node representation $\boldsymbol{h}_i^{(L)}$ after the $L$-th layer of the GNN model can be subsequently used for various downstream tasks (e.g., node classification and graph classification) with a trainable classifier $g$.

## 4 METHODOLOGY

In this section, we present our proposed method EdgePrompt and its variant EdgePrompt+. Figure 1 illustrates the difference between node prompt-based methods (Sun et al., 2023; Fang et al., 2023) and our edge prompt-based method. We first formulate the research problem studied in this paper. Then we introduce our design on edge prompts in EdgePrompt and EdgePrompt+ in detail. Furthermore, we provide comprehensive analyses to demonstrate that our method has the capability of benefiting pre-trained GNN models for node classification tasks. At last, we extend our method to graph classification as the downstream task.

### 4.1 PROBLEM SETTING

This study focuses on the standard problem of graph prompt tuning following previous studies (Fang et al., 2023; Sun et al., 2023). We consider a GNN model pre-trained by a pre-training task. We aim to adapt the pre-trained GNN model to a downstream task on a graph dataset through graph prompt tuning while keeping its parameters frozen. Specifically, given a pre-trained GNN model $f$, the goal is to transform the input graph $\mathcal{G}$ to a prompted graph $\mathcal{G}' = \mathcal{T}(\mathcal{G})$ with learnable prompts and obtain expressive node representations on $\mathcal{G}'$ by $f$ for a specific downstream task. Here, $\mathcal{T}$ is a graph transformation to obtain $\mathcal{G}'$ by adding prompts to $\mathcal{G}$. The key problem in graph prompt tuning is to design and learn suitable graph prompts to benefit downstream tasks.

### 4.2 EDGE PROMPT DESIGN

Inspired by pixel-level visual prompts (Bahng et al., 2022; Wu et al., 2022) in Computer Vision, the existing studies (Sun et al., 2023; Fang et al., 2023) design graph prompts at the data level by adding extra learnable prompt vectors to node features. Nevertheless, this strategy does not take account of the dependencies between nodes in graph data, which can significantly impact the final node representations via the message-passing mechanism in GNN models (Fatemi et al., 2021; Sun et al., 2022b; Liu et al., 2022a;b). Motivated by this, we propose to design our graph prompt tuning method from the perspective of edges in this study.

**EdgePrompt.** Considering the dependencies between nodes in graph data, we design learnable prompt vectors on edges and manipulate the input graph to a prompted one with the edge prompts; therefore, the pre-trained GNN model can generate expressive node representations on the prompted graph for the downstream task. More concretely, for each edge $(v_i, v_j) \in \mathcal{E}$, we aim to learn a prompt vector $\boldsymbol{e}_{ij}^{(l)} \in \mathbb{R}^{D_{l-1}}$ on it at the $l$-th layer of the pre-trained GNN model. Typically, this prompt vector can be regarded as the learnable properties of edges. As discussed previously, one critical challenge arises here: many popular GNN models, such as GCN (Kipf & Welling, 2017), do not accommodate edge attributes during the message-passing mechanism. Therefore, they are unable to absorb $\boldsymbol{e}_{ij}^{(l)}$ into node representations. To overcome this issue, we propose to aggregate the prompt vector along with node representations through the message-passing mechanism during

the forward pass at each layer of the pre-trained GNN model. Specifically, to compute $\boldsymbol{h}_i^{(l)}$ of each node $v_i$ at the $l$-th layer, the GNN model will aggregate not only $\boldsymbol{h}_j^{(l-1)}$ from its neighboring node $v_j \in \mathcal{N}(v_i)$ but also $\boldsymbol{e}_{ij}^{(l)}$ associated with edge $(v_i, v_j)$. Mathematically, we can reformulate Equation (1) with the edge prompt vector at the $l$-th layer of the pre-trained GNN model by

$$\boldsymbol{h}_i^{(l)} = \text{COMB}^{(l)}(\boldsymbol{h}_i^{(l-1)}, \text{AGG}^{(l)}(\{\boldsymbol{h}_j^{(l-1)} : v_j \in \mathcal{N}(v_i)\}, \{\boldsymbol{e}_{ij}^{(l)} : v_j \in \mathcal{N}(v_i)\})). \qquad (2)$$

To obtain the prompt vector, one simple yet effective way is to learn a global prompt vector shared by all the edges. Let $\boldsymbol{p}^{(l)} \in \mathbb{R}^{D_{l-1}}$ denote the global prompt vector at the $l$-th layer of the pre-trained GNN model. The prompt vector for each edge $(v_i, v_j)$ at the $l$-th layer can be written as

$$\boldsymbol{e}_{ij}^{(l)} = \boldsymbol{p}^{(l)}, \quad \forall (v_i, v_j) \in \mathcal{E}. \qquad (3)$$

The above design with global prompt vectors on edges is termed EdgePrompt in our method.

**EdgePrompt+.** Although EdgePrompt designs graph prompts from the perspective of edges, a single shared prompt vector for all the edges is insufficient to model different complex dependencies between nodes. Motivated by this, we conceive an advanced version of the above EdgePrompt, called EdgePrompt+, to learn customized prompt vectors on edges. Specifically, instead of using a shared prompt vector $\boldsymbol{p}^{(l)}$ for all the edges at the $l$-th layer, each edge $(v_i, v_j) \in \mathcal{E}$ will learn its own customized prompt vector $\boldsymbol{e}_{ij}^{(l)}$. Nevertheless, learning $|\mathcal{E}|$ independent prompt vectors is infeasible in practice. When we optimize prompt vectors for downstream tasks (e.g., node classification), we may have only a limited number of labeled nodes. Therefore, most edges cannot receive supervision information (Fatemi et al., 2021) for optimizing their prompt vectors, especially in a few-shot setting. In this case, it will be hard to directly learn $\boldsymbol{e}_{ij}^{(l)}$ for edge $(v_i, v_j) \in \mathcal{E}$ if it is not involved in computing the representations of any labeled nodes. To overcome this issue, we propose to learn the prompt vectors as a weighted average of multiple anchor prompts. To achieve this, we first construct a set of $M_l$ anchor prompts $\mathcal{P}^{(l)} = \{\boldsymbol{p}_1^{(l)}, \boldsymbol{p}_2^{(l)}, \cdots, \boldsymbol{p}_{M_l}^{(l)}\}$ at the $l$-th layer of the pre-trained GNN model, where each vector $\boldsymbol{p}_m^{(l)} \in \mathbb{R}^{D_{l-1}}$ is a learnable anchor prompt. For each edge $(v_i, v_j) \in \mathcal{E}$, its customized prompt vector $\boldsymbol{e}_{ij}^{(l)}$ at the $l$-th layer is computed as the weighted average of the anchor prompts in $\mathcal{P}^{(l)}$ with the score vector $\boldsymbol{b}_{ij}^{(l)} \in \mathbb{R}^{M_l}$. Mathematically, we can obtain $\boldsymbol{e}_{ij}^{(l)}$ at the $l$-th layer by

$$\boldsymbol{e}_{ij}^{(l)} = \sum_{m=1}^{M_l} b_{ijm}^{(l)} \cdot \boldsymbol{p}_m^{(l)}, \qquad (4)$$

where $b_{ijm}^{(l)}$ denotes the $m$-th entry in $\boldsymbol{b}_{ij}^{(l)}$. Since all the edges share the same anchor prompts $\mathcal{P}^{(l)}$ at the $l$-th layer, the score vector $\boldsymbol{b}_{ij}^{(l)}$ directly determines how $\boldsymbol{e}_{ij}^{(l)}$ differs from those of the other edges. Therefore, our next goal is to conceive an effective strategy to obtain the desired $\boldsymbol{b}_{ij}^{(l)}$. According to Equation (2), $\boldsymbol{e}_{ij}^{(l)}$ of edge $(v_i, v_j)$ affects message passing between nodes $v_i$ and $v_j$, so we may naturally consider $\boldsymbol{b}_{ij}^{(l)}$ to depend on both nodes $v_i$ and $v_j$. Motivated by this, we propose to compute $\boldsymbol{b}_{ij}^{(l)}$ at the $l$-th layer using a score function $\phi^{(l)}$ followed by the softmax operation. Formally, we compute $\boldsymbol{b}_{ij}^{(l)}$ by

$$\boldsymbol{b}_{ij}^{(l)} = \text{Softmax}(\phi^{(l)}(v_i, v_j)), \qquad (5)$$

where Softmax$(\cdot)$ represents the softmax operation. Here, $\phi^{(l)}$ takes each pair of nodes $v_i$ and $v_j$ as the input and generates the score vector. Basically, it describes the relationship of two nodes at the $l$-th layer and embeds them into a single vector. Many typical formulations (Veličković et al., 2018; Brody et al., 2022; Yang et al., 2021) can be used to achieve this goal. In this study, we adopt the classic attention mechanism (Veličković et al., 2018) as $\phi^{(l)}$ by

$$\phi^{(l)}(v_i, v_j) = \text{LeakyReLU}([\boldsymbol{h}_i^{(l-1)}||\boldsymbol{h}_j^{(l-1)}] \cdot \boldsymbol{W}^{(l)}), \qquad (6)$$

where $\boldsymbol{W}^{(l)} \in \mathbb{R}^{2D_{l-1} \times M_l}$ is the weight matrix of $\phi^{(l)}$ at the $l$-th layer, and $[\cdot||\cdot]$ denotes the vector concatenation. In-depth investigations into different variants of the score function $\phi$ will be reserved

for our future work. It is worth noting that GPF-plus (Fang et al., 2023) can be regarded as a special case of EdgePrompt+ with the score function as a linear mapping of $\boldsymbol{x}_i$.

With the learnable edge prompts, we can obtain more suitable node representations $\boldsymbol{h}_i^{(L)}$ for node $v_i \in \mathcal{V}$ by the pre-trained GNN model for node classification. Given the labeled node set $\mathcal{V}_L \in \mathcal{V}$, we optimize our edge prompts and a classifier $g$ by

$$\min_{g,\{\mathcal{P}^{(1)},\cdots,\mathcal{P}^{(L)},\boldsymbol{W}^{(1)},\cdots,\boldsymbol{W}^{(L)}\}} \frac{1}{|\mathcal{V}_L|} \sum_{v_i \in \mathcal{V}_L} \ell_D(g(f(\mathcal{G}')_i), y_i), \tag{7}$$

where $y_i$ is the ground-truth label of node $v_i \in \mathcal{V}_L$, and $\ell_D$ is the downstream task loss, i.e., the cross-entropy loss for classification tasks.

## 4.3 ANALYSIS OF EDGE PROMPT TUNING FOR NODE CLASSIFICATION

In this subsection, we provide a comprehensive analysis to investigate why our proposed Edge-Prompt+ is more effective for node classification than existing approaches, particularly those that focus on learning additional prompt vectors on node features.

We first provide our insights regarding the issue of uniform message passing on prompt vectors. As introduced previously, GPF-plus (Fang et al., 2023) and All-in-one (Sun et al., 2023) design learnable prompt vectors on the node level and manipulate the input graph by adding the prompt vectors to node features. For each node $v_i$, its learned prompt vector $\boldsymbol{p}_i$ completely depends on its node feature $\boldsymbol{x}_i$. In many prevalent GNN models, such as GCNs, the prompt vector will be uniformly aggregated by neighboring nodes through the message-passing mechanism (Yang et al., 2021). Taking node $v_1$ in Figure 1(a) as an example, its two neighboring nodes $v_2$ and $v_3$ will always receive the same prompt vector $\boldsymbol{p}_1$ from node $v_1$ in pre-trained GCN models. Unfortunately, such propagation of prompt vectors may not benefit node classification. Instead, the prompt vector aggregated by a node can retain adverse information from different classes. In contrast, our proposed EdgePrompt+ designs prompt vectors on edges. Unlike one shared prompt vector of a node for all its neighboring nodes, EdgePrompt+ enables these neighboring nodes to receive different learned prompt vectors (e.g., $\boldsymbol{e}_{21}$ and $\boldsymbol{e}_{31}$ in Figure 1(b)) from the node. In this way, the issue of uniform passing on prompt vectors can be mitigated.

Furthermore, we would like to provide a theoretical analysis of how edge prompts in our proposed EdgePrompt+ can benefit node classification. Our analysis is based on random graphs generated by the contextual stochastic block model (CSBM) (Tsitsulin et al., 2022; Ma et al., 2022). Specifically, we consider a random graph $\mathcal{G}$ generated by the CSBM consisting of two node classes $c_1$ and $c_2$. For each node $v_i$, its node feature $\boldsymbol{x}_i$ follows a Gaussian distribution $\boldsymbol{x}_i \sim \mathcal{N}(\boldsymbol{\mu}_1, \boldsymbol{I})$ if it is from class $c_1$, otherwise $\boldsymbol{x}_i \sim \mathcal{N}(\boldsymbol{\mu}_2, \boldsymbol{I})$. Generally, we assume $\boldsymbol{\mu}_1 \neq \boldsymbol{\mu}_2$. In the graph $\mathcal{G}$, edges are generated following an intra-class probability $p$ and an inter-class probability $q$. More concretely, a pair of nodes will be connected by an edge with probability $p$ if they are from the same class; otherwise, the probability is $q$. In this section, we use $\mathcal{G} \sim \text{CSBM}(\boldsymbol{\mu}_1, \boldsymbol{\mu}_2, p, q)$ to denote a random graph generated by the CSBM.

Our analysis aims to investigate the improvement of linear separability under pre-trained GCN models caused by edge prompts in EdgePrompt+. Specifically, we focus on the linear classifiers with the largest margin based on node representations after GCN operations with and without edge prompts. Typically, if the expected distance between the two class centroids is larger, the node representations will have higher linear separability by the linear classifier.

**Theorem 1.** *Given a random graph $\mathcal{G} \sim CSBM(\boldsymbol{\mu}_1, \boldsymbol{\mu}_2, p, q)$ and a pre-trained GCN model $f$, there always exist a set of $M \geq 2$ anchor prompts $\mathcal{P} = \{\boldsymbol{p}_1, \boldsymbol{p}_2, \cdots, \boldsymbol{p}_M\}$ and the score vectors $\boldsymbol{b}_{i,j}$ for each edge $(v_i, v_j)$ that improve the expected distance after GCN operation between classes $c_1$ and $c_2$ to $T$ times without using edge prompts, where $T \in (1, 1 + \frac{p}{|p-q|}]$.*

A detailed proof can be found in Appendix A. According to Theorem 1, we will have a larger expected distance between the two class centroids after GCN operation with edge prompts in Edge-Prompt+. In this case, the node representations from the two classes will have a lower probability of being misclassified. Therefore, we can conclude that our proposed EdgePrompt+ benefits pre-trained GNN models for node classification.

### 4.4 EXTENSION TO GRAPH CLASSIFICATION

In the last subsection, we present our edge prompt design in EdgePrompt and EdgePrompt+ for node classification. As discussed previously, edge prompts should be capable of handling various downstream tasks, including graph classification. In this subsection, we would like to introduce how EdgePrompt and EdgePrompt+ tackle graph classification.

In graph classification, we have a set of labeled graphs $\{\mathcal{G}_1, \mathcal{G}_2, \cdots, \mathcal{G}_K\}$ with their label set $\{y_1, y_2, \cdots, y_K\}$. To obtain the representation of the entire graph $\mathcal{G}$, we typically integrate the final representations of all nodes in $\mathcal{G}$ via a permutation-invariant READOUT function (Xu et al., 2019), such as *sum* and *mean*, as the entire graph's representation $\boldsymbol{h}_{\mathcal{G}} = \text{READOUT}(\{\boldsymbol{h}_i | v_i \in \mathcal{V}\})$. Therefore, we can optimize our edge prompts and a classifier $g$ by

$$\min_{g, \{\mathcal{P}^{(1)}, \cdots, \mathcal{P}^{(L)}, \boldsymbol{W}^{(1)}, \cdots, \boldsymbol{W}^{(L)}\}} \frac{1}{K} \sum_{k=1}^{K} \ell_D(g(f(\mathcal{G}'_k)), y_k). \tag{8}$$

Now we analyze the capability of EdgePrompt for graph classifications. The goal of our analysis is to investigate whether learning edge prompts in EdgePrompt can result in consistent graph representations with those using any prompt strategies. To this end, we propose the following theorem.

**Theorem 2.** *Given an input graph $\mathcal{G} = (\boldsymbol{X}, \boldsymbol{A})$ and its transformation $\mathcal{G}' = (\boldsymbol{X}', \boldsymbol{A}')$ by an arbitrary transformation function $\mathcal{T}$, there exists a set of edge prompt vectors $\{\boldsymbol{p}^{(1)}, \boldsymbol{p}^{(2)}, \cdots, \boldsymbol{p}^{(L)}\}$ in EdgePrompt that can satisfy*

$$f(\boldsymbol{X}, \boldsymbol{A}, \{\boldsymbol{p}^{(1)}, \cdots, \boldsymbol{p}^{(L)}\}) = f(\boldsymbol{X}', \boldsymbol{A}') \tag{9}$$

*for any pre-trained GNN model $f$.*

The complete proof of Theorem 2 is provided in Appendix B. According to Theorem 2, we can conclude that our edge prompts have the capability to get graph $\mathcal{G}$'s representation which is equal to those of its variants by transformations with any prompt strategies. According to Theorem 1 by Fang et al. (2023), our EdgePrompt has a comparable universal capability with GPF. Since EdgePrompt+ provides finer edge prompts than EdgePrompt, it will have a stronger universality than EdgePrompt.

## 5 EXPERIMENTS

### 5.1 EXPERIMENTAL SETUP

**Datasets and downstream tasks.** We evaluate the effectiveness of our proposed method on node classification over five public graph datasets, including Cora (Yang et al., 2016), CiteSeer (Yang et al., 2016), Pubmed (Yang et al., 2016), ogbn-arxiv (Hu et al., 2020a), and Flickr (Zeng et al., 2020). In addition, we adopt five graph datasets from TUDataset (Morris et al., 2020), including ENZYMES, DD, NCI1, NCI109, and Mutagenicity, to conduct experiments for graph classification. Basic information and statistics about these datasets can be found in Appendix C.1.

**Pre-training strategies.** To evaluate the compatibility of our proposed method with various pre-training strategies, we consider four pre-training strategies in our experiments. For contrastive methods, we use GraphCL (You et al., 2020) and SimGRACE (Xia et al., 2022). For generative methods, we use two edge prediction-based methods, i.e., EP-GPPT and EP-GraphPrompt, adopted by GPPT (Sun et al., 2022a) and GraphPrompt (Liu et al., 2023), respectively. We provide detailed descriptions of these pre-training strategies in Appendix C.2.

**Baselines.** We evaluate our proposed method against five state-of-the-art graph prompt tuning methods in our experiments, including GPPT (Sun et al., 2022a), GraphPrompt (Liu et al., 2023) All-in-one (Sun et al., 2023), GPF (Fang et al., 2023), and GPF-plus (Fang et al., 2023). Since GPPT is specifically designed for node classification, we only report its performance for node classification tasks. In addition, we also report the performance of solely training classifiers without any prompts (named as *Classifier Only*) in our experiments.

**Implementation details.** In our experiments, We use a 2-layer GCN (Kipf & Welling, 2017) as the backbone for node classification tasks and a 5-layer GIN (Xu et al., 2019) as the backbone for graph classification tasks. The size of hidden layers is set as 128. The classifier adopted for downstream

Table 2: Accuracy on 5-shot node classification tasks over five datasets. The best-performing method is **bolded** and the runner-up is underlined.

| Pre-training Strategies | Tuning Methods | Cora | CiteSeer | Pubmed | ogbn-arxiv | Flickr |
|---|---|---|---|---|---|---|
| GraphCL | Classifier Only | $53.05_{\pm4.76}$ | $38.62_{\pm3.43}$ | $64.28_{\pm4.51}$ | $21.15_{\pm1.64}$ | $24.32_{\pm2.93}$ |
| | GPPT | $50.96_{\pm6.67}$ | $39.50_{\pm1.67}$ | $60.47_{\pm4.75}$ | $17.99_{\pm1.14}$ | $24.35_{\pm1.84}$ |
| | GraphPrompt | $55.71_{\pm4.62}$ | $40.81_{\pm2.11}$ | $63.47_{\pm2.23}$ | $21.03_{\pm1.92}$ | $\mathbf{26.08_{\pm3.44}}$ |
| | ALL-in-one | $38.00_{\pm4.17}$ | $40.27_{\pm2.09}$ | $58.61_{\pm3.49}$ | $16.42_{\pm2.98}$ | $25.08_{\pm3.44}$ |
| | GPF | $58.52_{\pm4.07}$ | $43.55_{\pm2.80}$ | $67.67_{\pm3.14}$ | $21.73_{\pm1.75}$ | $23.98_{\pm1.71}$ |
| | GPF-plus | $52.24_{\pm4.59}$ | $38.47_{\pm3.27}$ | $64.30_{\pm4.58}$ | $21.03_{\pm1.96}$ | $25.32_{\pm2.02}$ |
| | EdgePrompt | $58.60_{\pm4.46}$ | $43.31_{\pm3.23}$ | $\mathbf{67.76_{\pm3.01}}$ | $21.90_{\pm1.71}$ | $24.83_{\pm2.78}$ |
| | EdgePrompt+ | $\mathbf{62.88_{\pm6.43}}$ | $\mathbf{46.20_{\pm0.99}}$ | $\underline{67.41_{\pm5.25}}$ | $\mathbf{23.18_{\pm1.26}}$ | $\underline{25.57_{\pm3.04}}$ |
| SimGRACE | Classifier Only | $52.27_{\pm2.74}$ | $40.45_{\pm3.55}$ | $56.72_{\pm3.80}$ | $20.75_{\pm2.92}$ | $25.53_{\pm3.98}$ |
| | GPPT | $52.07_{\pm7.65}$ | $40.25_{\pm3.29}$ | $58.65_{\pm5.12}$ | $17.76_{\pm1.80}$ | $23.37_{\pm4.66}$ |
| | GraphPrompt | $51.42_{\pm2.80}$ | $41.74_{\pm2.22}$ | $55.98_{\pm2.94}$ | $20.48_{\pm2.57}$ | $25.88_{\pm3.81}$ |
| | ALL-in-one | $34.64_{\pm4.06}$ | $38.95_{\pm2.35}$ | $54.18_{\pm4.70}$ | $16.72_{\pm2.90}$ | $27.68_{\pm4.58}$ |
| | GPF | $58.23_{\pm4.19}$ | $44.87_{\pm4.35}$ | $61.55_{\pm3.79}$ | $21.86_{\pm2.91}$ | $26.51_{\pm4.69}$ |
| | GPF-plus | $52.27_{\pm3.34}$ | $\underline{41.02_{\pm3.49}}$ | $56.95_{\pm3.86}$ | $21.44_{\pm3.77}$ | $28.35_{\pm5.50}$ |
| | EdgePrompt | $\underline{58.37_{\pm4.51}}$ | $43.94_{\pm4.15}$ | $\underline{61.10_{\pm3.69}}$ | $21.85_{\pm2.54}$ | $\mathbf{30.12_{\pm5.04}}$ |
| | EdgePrompt+ | $\mathbf{62.40_{\pm7.97}}$ | $\mathbf{46.62_{\pm2.53}}$ | $\mathbf{64.91_{\pm5.58}}$ | $\mathbf{22.74_{\pm2.34}}$ | $\underline{28.50_{\pm4.08}}$ |
| EP-GPPT | Classifier Only | $28.65_{\pm4.82}$ | $26.77_{\pm2.03}$ | $40.14_{\pm5.69}$ | $11.57_{\pm1.91}$ | $28.39_{\pm7.44}$ |
| | GPPT | $\underline{41.28_{\pm6.92}}$ | $\underline{35.32_{\pm1.27}}$ | $\underline{53.41_{\pm3.99}}$ | $13.73_{\pm1.16}$ | $29.83_{\pm3.73}$ |
| | GraphPrompt | $31.65_{\pm3.33}$ | $26.98_{\pm1.24}$ | $44.18_{\pm5.57}$ | $11.31_{\pm1.89}$ | $26.02_{\pm1.16}$ |
| | ALL-in-one | $31.57_{\pm2.16}$ | $28.87_{\pm2.57}$ | $46.02_{\pm4.23}$ | $15.94_{\pm0.75}$ | $\underline{31.89_{\pm1.14}}$ |
| | GPF | $37.56_{\pm3.81}$ | $29.74_{\pm1.73}$ | $48.86_{\pm7.32}$ | $16.95_{\pm1.58}$ | $29.68_{\pm6.73}$ |
| | GPF-plus | $28.87_{\pm3.18}$ | $26.65_{\pm1.91}$ | $40.32_{\pm5.77}$ | $11.78_{\pm1.55}$ | $29.41_{\pm6.79}$ |
| | EdgePrompt | $37.26_{\pm4.53}$ | $29.83_{\pm1.01}$ | $47.20_{\pm7.06}$ | $\underline{17.22_{\pm1.31}}$ | $31.17_{\pm6.58}$ |
| | EdgePrompt+ | $\mathbf{56.41_{\pm3.62}}$ | $\mathbf{43.49_{\pm2.62}}$ | $\mathbf{61.51_{\pm4.91}}$ | $\mathbf{17.78_{\pm2.16}}$ | $\mathbf{32.70_{\pm6.21}}$ |
| EP-GraphPrompt | Classifier Only | $59.00_{\pm5.74}$ | $44.54_{\pm4.44}$ | $72.09_{\pm5.70}$ | $31.28_{\pm1.50}$ | $27.83_{\pm4.77}$ |
| | GPPT | $54.29_{\pm7.90}$ | $45.81_{\pm3.54}$ | $66.56_{\pm4.06}$ | $25.34_{\pm1.85}$ | $28.41_{\pm3.68}$ |
| | GraphPrompt | $60.22_{\pm4.04}$ | $47.07_{\pm3.09}$ | $73.13_{\pm5.07}$ | $\underline{32.40_{\pm1.30}}$ | $28.10_{\pm3.27}$ |
| | ALL-in-one | $42.55_{\pm2.99}$ | $44.36_{\pm2.52}$ | $67.66_{\pm6.38}$ | $15.22_{\pm3.04}$ | $31.79_{\pm6.19}$ |
| | GPF | $62.62_{\pm6.40}$ | $49.02_{\pm4.53}$ | $73.62_{\pm6.42}$ | $31.88_{\pm1.08}$ | $28.98_{\pm5.30}$ |
| | GPF-plus | $58.23_{\pm5.68}$ | $44.60_{\pm4.47}$ | $72.15_{\pm5.64}$ | $31.58_{\pm1.09}$ | $28.96_{\pm4.63}$ |
| | EdgePrompt | $\underline{62.74_{\pm6.77}}$ | $48.69_{\pm4.36}$ | $73.60_{\pm5.14}$ | $\mathbf{32.67_{\pm1.83}}$ | $29.81_{\pm3.59}$ |
| | EdgePrompt+ | $\mathbf{64.47_{\pm7.04}}$ | $\mathbf{49.71_{\pm2.25}}$ | $\mathbf{73.72_{\pm5.10}}$ | $31.41_{\pm1.88}$ | $\mathbf{32.09_{\pm4.93}}$ |

tasks is linear probes for all the methods. We use an Adam optimizer (Kingma & Ba, 2015) with learning rates 0.001 for all the methods. The batch size is set as 32. The number of epochs is set to 200 for graph prompt tuning. The default number of anchor prompts at each GNN layer is 10 for node classification tasks and 5 for graph classification tasks. We use the 5-shot setting for node classification tasks and the 50-shot setting for graph classification tasks. We conduct experiments five times with different random seeds and report the average results in our experiments.

## 5.2 MAIN RESULTS

We first compare the overall performance of our proposed methods and other baselines. Table 2 reports the results of our method and six baselines on 5-shot node classification tasks over five datasets under four pre-training strategies. According to the table, we observe that our method can consistently achieve the best or most competitive performance among graph prompt tuning methods across different pre-training strategies. Generally, EdgePrompt+ has better performance than EdgePrompt, which is consistent with our analyses in Section 4.3 and validates the necessity of our design on customized edge prompts.

In addition, we conduct experiments on 50-shot graph classification tasks over five datasets under four pre-training strategies and report the results in Table 3. According to the table, we observe that EdgePrompt+ can always get the best place or runner-up for every experimental setting, espe-

Table 3: Accuracy on 50-shot graph classification tasks over five datasets. The best-performing method is **bolded** and the runner-up underlined.

| Pre-training Strategies | Tuning Methods | ENZYMES | DD | NCI1 | NCI109 | Mutagenicity |
|---|---|---|---|---|---|---|
| GraphCL | Classifier Only | $30.50_{\pm1.16}$ | $62.89_{\pm2.19}$ | $62.49_{\pm1.95}$ | $61.68_{\pm0.93}$ | $66.62_{\pm1.87}$ |
| | GraphPrompt | $27.83_{\pm1.61}$ | $64.33_{\pm1.79}$ | $63.19_{\pm1.71}$ | $62.18_{\pm0.48}$ | $\mathbf{67.62_{\pm0.65}}$ |
| | ALL-in-one | $25.92_{\pm0.55}$ | $66.54_{\pm1.82}$ | $57.52_{\pm2.61}$ | $62.74_{\pm0.78}$ | $63.43_{\pm2.53}$ |
| | GPF | $30.08_{\pm1.25}$ | $64.54_{\pm2.22}$ | $62.66_{\pm1.83}$ | $62.29_{\pm0.90}$ | $66.54_{\pm1.85}$ |
| | GPF-plus | $31.00_{\pm1.50}$ | $67.26_{\pm2.29}$ | $64.56_{\pm1.10}$ | $62.84_{\pm0.22}$ | $66.82_{\pm1.63}$ |
| | EdgePrompt | $29.50_{\pm1.57}$ | $64.16_{\pm2.13}$ | $63.05_{\pm2.11}$ | $62.59_{\pm0.93}$ | $66.87_{\pm1.88}$ |
| | EdgePrompt+ | $\mathbf{34.00_{\pm1.25}}$ | $\mathbf{67.98_{\pm2.05}}$ | $\mathbf{66.30_{\pm2.54}}$ | $\mathbf{66.52_{\pm0.91}}$ | $\underline{67.47_{\pm2.37}}$ |
| SimGRACE | Classifier Only | $27.07_{\pm1.04}$ | $61.77_{\pm2.40}$ | $61.27_{\pm3.64}$ | $62.12_{\pm1.10}$ | $67.36_{\pm0.71}$ |
| | GraphPrompt | $26.87_{\pm1.47}$ | $62.58_{\pm1.84}$ | $\underline{62.45_{\pm1.52}}$ | $62.41_{\pm0.69}$ | $\underline{68.03_{\pm0.78}}$ |
| | ALL-in-one | $25.73_{\pm1.18}$ | $65.16_{\pm1.47}$ | $58.52_{\pm1.59}$ | $62.01_{\pm0.66}$ | $64.43_{\pm1.00}$ |
| | GPF | $28.53_{\pm1.76}$ | $65.64_{\pm0.70}$ | $61.45_{\pm3.13}$ | $61.90_{\pm1.26}$ | $67.19_{\pm0.74}$ |
| | GPF-plus | $27.33_{\pm2.01}$ | $\underline{67.20_{\pm1.56}}$ | $61.61_{\pm2.89}$ | $62.84_{\pm0.23}$ | $67.69_{\pm0.64}$ |
| | EdgePrompt | $29.33_{\pm2.30}$ | $63.97_{\pm2.14}$ | $62.02_{\pm3.02}$ | $62.02_{\pm1.03}$ | $67.55_{\pm0.85}$ |
| | EdgePrompt+ | $\mathbf{32.67_{\pm2.53}}$ | $\mathbf{67.72_{\pm1.62}}$ | $\mathbf{67.07_{\pm1.96}}$ | $\mathbf{66.53_{\pm1.30}}$ | $\mathbf{68.31_{\pm1.36}}$ |
| EP-GPPT | Classifier Only | $29.08_{\pm1.35}$ | $62.12_{\pm2.82}$ | $56.85_{\pm4.35}$ | $62.27_{\pm0.78}$ | $66.30_{\pm1.78}$ |
| | GraphPrompt | $26.67_{\pm1.60}$ | $61.61_{\pm1.91}$ | $58.77_{\pm0.97}$ | $62.16_{\pm0.89}$ | $66.37_{\pm1.17}$ |
| | ALL-in-one | $24.92_{\pm1.33}$ | $63.61_{\pm2.12}$ | $59.14_{\pm2.12}$ | $59.70_{\pm1.37}$ | $64.86_{\pm1.60}$ |
| | GPF | $28.33_{\pm1.73}$ | $63.48_{\pm2.08}$ | $58.14_{\pm4.16}$ | $62.52_{\pm1.39}$ | $66.10_{\pm0.96}$ |
| | GPF-plus | $\underline{29.25_{\pm1.30}}$ | $\mathbf{66.92_{\pm2.34}}$ | $62.93_{\pm3.23}$ | $64.13_{\pm1.42}$ | $\underline{67.57_{\pm1.45}}$ |
| | EdgePrompt | $28.33_{\pm3.41}$ | $64.03_{\pm2.26}$ | $59.85_{\pm3.15}$ | $62.98_{\pm1.44}$ | $66.36_{\pm1.22}$ |
| | EdgePrompt+ | $\mathbf{32.75_{\pm2.26}}$ | $\underline{66.16_{\pm1.60}}$ | $\mathbf{63.58_{\pm2.07}}$ | $\mathbf{65.15_{\pm1.60}}$ | $\mathbf{68.35_{\pm1.57}}$ |
| EP-GraphPrompt | Classifier Only | $31.33_{\pm3.22}$ | $62.58_{\pm2.40}$ | $62.09_{\pm2.31}$ | $60.19_{\pm1.71}$ | $65.13_{\pm0.81}$ |
| | GraphPrompt | $30.20_{\pm1.93}$ | $64.72_{\pm1.98}$ | $62.57_{\pm1.45}$ | $62.32_{\pm0.95}$ | $\underline{65.85_{\pm0.65}}$ |
| | ALL-in-one | $29.07_{\pm1.16}$ | $65.60_{\pm2.38}$ | $58.67_{\pm2.42}$ | $57.69_{\pm1.08}$ | $64.66_{\pm0.76}$ |
| | GPF | $\underline{30.93_{\pm1.76}}$ | $66.21_{\pm1.66}$ | $61.80_{\pm2.78}$ | $62.27_{\pm1.18}$ | $65.61_{\pm0.59}$ |
| | GPF-plus | $30.67_{\pm3.06}$ | $\mathbf{67.50_{\pm2.45}}$ | $62.59_{\pm2.09}$ | $61.98_{\pm1.60}$ | $65.51_{\pm1.10}$ |
| | EdgePrompt | $30.80_{\pm2.09}$ | $65.87_{\pm1.35}$ | $61.75_{\pm2.49}$ | $62.33_{\pm1.65}$ | $65.77_{\pm0.90}$ |
| | EdgePrompt+ | $\mathbf{33.27_{\pm2.71}}$ | $\underline{67.47_{\pm2.14}}$ | $\mathbf{65.06_{\pm1.84}}$ | $\mathbf{64.64_{\pm1.57}}$ | $\mathbf{66.42_{\pm1.31}}$ |

cially over ENZYMES, NCI1, and NCI109. In addition, we observe that GPF and EdgePrompt have relatively small performance gaps (always $< 1.8\%$) in the table (we also observe this in node classification tasks). As indicated in Theorem 2, our proposed EdgePrompt has a comparable universal capability with GPF to achieve graph representations equivalent to any graph transformation. These observations effectively support our theoretical claim in this study.

## 5.3 CONVERGENCE ANALYSIS

In this subsection, we would like to investigate the convergence speeds of our method compared with baselines. Figure 2 illustrates the accuracy curves of our method and the baselines under two pre-training strategies. According to Figure 2, we can observe that EdgePrompt+ can generally converge faster than other methods.

## 5.4 INFLUENCE OF PROMPT NUMBERS

We conduct experiments to investigate the impact of different numbers of anchor prompts on model utility. Figure 3 and Figure 4 illustrate the performance of EdgePrompt+ with 1, 5, 10, 20, and 50 anchor prompts at each layer for node classification and graph classification tasks, respectively. Note that EdgePrompt+ will be degraded to EdgePrompt when we have only one anchor prompt at each GNN layer. From the two figures, we can conclude only one anchor prompt vector (i.e., EdgePrompt) is insufficient in most cases where each edge will learn a global prompt vector. In the meantime, EdgePrompt+ with too many anchor prompts (e.g., 50) may not further improve the performance. We recommend 5 or 10 as the initial number of anchor prompts.

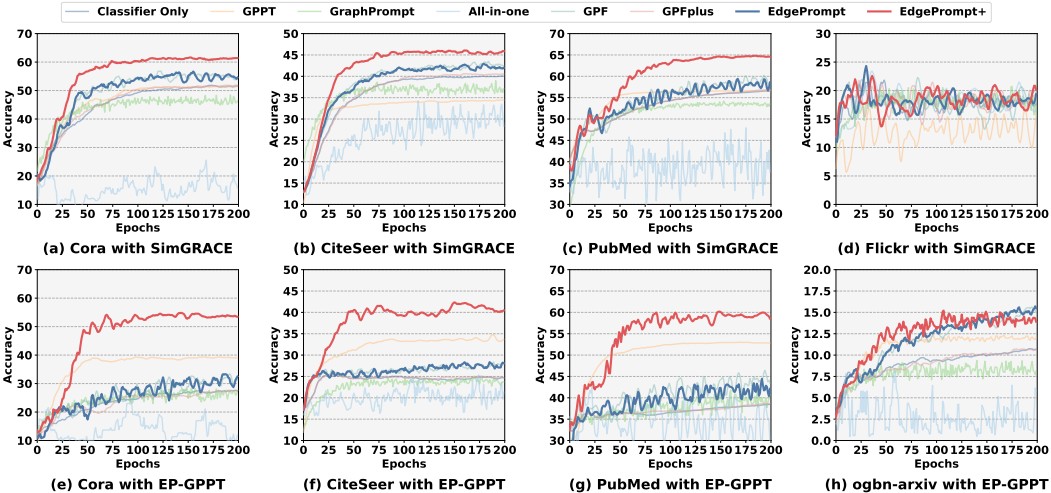

Figure 2: Convergence speeds of different methods.

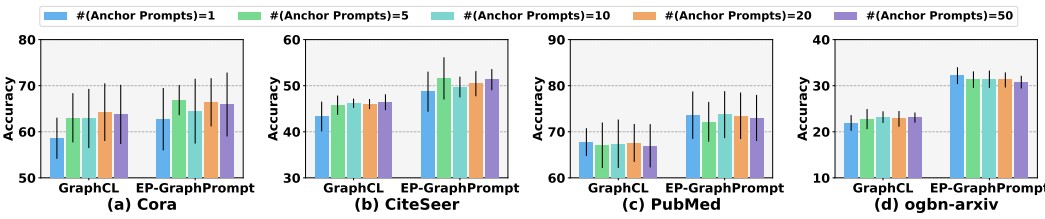

Figure 3: Results of EdgePrompt+ with varying numbers of anchor prompts on node classification.

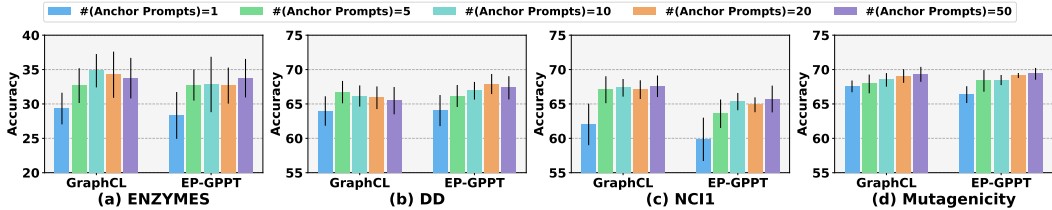

Figure 4: Results of EdgePrompt+ with varying numbers of anchor prompts on graph classification.

## 6 CONCLUSION

Graph prompt tuning is an emerging technique to bridge the objective gap between pre-training and downstream tasks. Unlike previous studies focusing on designing prompts on nodes, we propose a simple yet effective method, EdgePrompt and its variant EdgePrompt+, that manipulates the input graph by adding extra learnable prompt vectors to edges and thereby obtaining a prompted graph suitable for downstream tasks. We provide comprehensive theoretical analyses of our method regarding its capability of handling node classification and graph classification. We conduct extensive experiments over ten graph datasets under four pre-training strategies. Experiment results demonstrate the superiority of our method compared with six baselines.

### ACKNOWLEDGMENTS

This work is supported in part by the National Science Foundation under grants IIS-2006844, IIS-2144209, IIS-2223769, IIS-2331315, CNS-2154962, BCS-2228534, and CMMI-2411248, the Commonwealth Cyber Initiative Awards under grants VV-1Q24-011, VV-1Q25-004, and the iPRIME Fellowship Awards.

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

## A    PROOF OF THEOREM 1

**Theorem 1.** *Given a random graph $\mathcal{G} \sim CSBM(\boldsymbol{\mu}_1, \boldsymbol{\mu}_2, p, q)$ and a pre-trained GCN model $f$, there always exist a set of $M \geq 2$ anchor prompts $\mathcal{P} = \{\boldsymbol{p}_1, \boldsymbol{p}_2, \cdots, \boldsymbol{p}_M\}$ and the score vectors $\boldsymbol{b}_{i,j}$ for each edge $(v_i, v_j)$ that improve the expected distance after GCN operation between classes $c_1$ and $c_2$ to $T$ times without using edge prompts, where $T \in (1, 1 + \frac{p}{|p-q|}]$.*

*Proof.* For each node $v_i$ in graph $\mathcal{G}$, we can approximately regard that the labels of its neighboring nodes are independently sampled from a neighborhood distribution $\mathcal{D}_{c_1} = [\frac{p}{p+q}, \frac{q}{p+q}]$ if node $v_i$ is in class $c_1$ or $\mathcal{D}_{c_2} = [\frac{q}{p+q}, \frac{p}{p+q}]$ if node $v_i$ is in class $c_2$ (Ma et al., 2022). When we do not consider edge prompts, the expected feature obtained from the GCN operation will be

$$\mathbb{E}[\boldsymbol{h}_1] = \frac{p}{p + q} \cdot \boldsymbol{\mu}_1 + \frac{q}{p + q} \cdot \boldsymbol{\mu}_2 \tag{10}$$

for nodes in class $c_1$ and

$$\mathbb{E}[\boldsymbol{h}_2] = \frac{q}{p + q} \cdot \boldsymbol{\mu}_1 + \frac{p}{p + q} \cdot \boldsymbol{\mu}_2 \tag{11}$$

for nodes in class $c_2$. Here, we ignore the linear transformation in the GCN operation since it can be absorbed by the linear classifier. To evaluate the linear separability of linear classifiers, we calculate the expected distance $d$ between the two classes $c_1$ and $c_2$ by

$$d = ||\mathbb{E}[\boldsymbol{h}_1] - \mathbb{E}[\boldsymbol{h}_2]|| = \frac{|p - q|}{p + q} \cdot ||\boldsymbol{\mu}_1 - \boldsymbol{\mu}_2||. \tag{12}$$

When we consider edge prompts in EdgePrompt+, we need to involve them into the aggregation in the GCN operation. Without loss of generality, we can fix $b_{ijm} = 0$ for $m \in [3, M]$. Therefore, for each edge $(v_i, v_j)$, its prompt vector will be

$$\boldsymbol{e}_{ij} = \sum_{m=1}^{M_l} b_{ijm} \cdot \boldsymbol{p}_m = b_{ij1} \cdot \boldsymbol{p}_1 + b_{ij2} \cdot \boldsymbol{p}_2. \tag{13}$$

Obviously, $b_{ij2} = 1 - b_{ij1}$. In addition, we can set the two prompt vectors as $\boldsymbol{\mu}_1$ and $\boldsymbol{\mu}_2$, i.e.,

$$\boldsymbol{e}_{ij} = b_{ij1} \cdot \boldsymbol{\mu}_1 + b_{ij2} \cdot \boldsymbol{\mu}_2. \tag{14}$$

Then the new expected feature obtained from the GCN operation with edge prompts will be

$$\mathbb{E}[\boldsymbol{h}_1'] = \frac{p \cdot (\boldsymbol{\mu}_1 + b_{11} \cdot \boldsymbol{\mu}_1 + (1 - b_{11} \cdot \boldsymbol{\mu}_2)) + q \cdot (\boldsymbol{\mu}_2 + b_{12} \cdot \boldsymbol{\mu}_1 + (1 - b_{12} \cdot \boldsymbol{\mu}_2))}{p + q} \tag{15}$$

for nodes in class $c_1$ and

$$\mathbb{E}[\boldsymbol{h}_2'] = \frac{q \cdot (\boldsymbol{\mu}_1 + b_{21} \cdot \boldsymbol{\mu}_1 + (1 - b_{21} \cdot \boldsymbol{\mu}_2)) + p \cdot (\boldsymbol{\mu}_2 + b_{22} \cdot \boldsymbol{\mu}_1 + (1 - b_{22} \cdot \boldsymbol{\mu}_2))}{p + q} \tag{16}$$

for nodes in class $c_2$. Here, $b_{11} \in [0, 1]$ represents the expected score between nodes from class 1, $b_{22} \in [0, 1]$ represents the expected score between nodes from class 2, $b_{12} \in [0, 1]$ and $b_{21} \in [0, 1]$ represents the expected score between nodes across classes. Different from the original design in our method, we can set $b_{12} = b_{21}$ for simplicity. Therefore, the new expected distance with edge prompts will be

$$\begin{aligned}
d' &= \|\mathbb{E}[\mathbf{h}_1'] - \mathbb{E}[\mathbf{h}_2']\| \\
&= \left\| \frac{(p - q + b_{11} \cdot p - b_{22} \cdot p)\mu_1 - (-(1 - b_{11}) \cdot p - q + p + (1 - b_{22}) \cdot p)\mu_2}{p + q} \right\| \\
&= \frac{|(p - q + (b_{11} - b_{22}) \cdot p)|}{p + q} \cdot \|\mu_1 - \mu_2\|
\end{aligned} \tag{17}$$

To improve the linear separability of the two classes, we hope to get $d' > d$. In this case, we may assume

$$d' = T \cdot d = \frac{|T \cdot (p - q)|}{p + q} \cdot \|\mu_1 - \mu_2\| \tag{18}$$

with $T > 1$. Therefore, we need

$$b_{11} - b_{22} = \frac{(T - 1) \cdot (p - q)}{p}. \tag{19}$$

Since $b_{11} \in [0, 1]$ and $b_{22} \in [0, 1]$, we need

$$-1 \le \frac{(T - 1) \cdot (p - q)}{p} \le 1. \tag{20}$$

Then we have

$$T \le 1 + \frac{p}{|p - q|}. \tag{21}$$

Therefore, We can conclude that we can always find a set of $M \ge 2$ anchor prompts $\mathcal{P} = \{\boldsymbol{\mu}_1, \boldsymbol{\mu}_2, \boldsymbol{p}_3, \cdots, \boldsymbol{p}_M\}$ and the above score values for each edge $(v_i, v_j)$ that improve the expected distance after GCN operation between classes $c_1$ and $c_2$ to $T$ times without using edge prompts, where $T \in (1, 1 + \frac{p}{|p-q|}]$. □

## B  PROOF OF THEOREM 2

Before we prove Theorem 2, we would like to prove the following lemma.

**Lemma 1.** *Given an input graph $\mathcal{G} = (\boldsymbol{X}, \boldsymbol{A})$ and an extra feature prompt $\hat{\boldsymbol{p}}$ in GPF, there exists a set of edge prompt vectors $\{\boldsymbol{p}^{(1)}, \boldsymbol{p}^{(2)}, \cdots, \boldsymbol{p}^{(L)}\}$ in EdgePrompt that can satisfy*

$$f(\boldsymbol{X}, \boldsymbol{A}, \{\boldsymbol{p}^{(1)}, \cdots, \boldsymbol{p}^{(L)}\}) = f(\boldsymbol{X} + \hat{\boldsymbol{p}}, \boldsymbol{A}) \tag{22}$$

*for any pre-trained GNN model $f$.*

*Proof.* Following GPF (Fang et al., 2023), we first consider a single-layer GIN (Xu et al., 2019) with a linear transformation. Mathematically, we can compute the node representation matrix in a GIN layer by

$$\boldsymbol{H} = (\boldsymbol{A} + (1+\epsilon) \cdot \boldsymbol{I}) \cdot \boldsymbol{X} \cdot \boldsymbol{W} = \boldsymbol{A} \cdot \boldsymbol{X} \cdot \boldsymbol{W} + (1+\epsilon) \cdot \boldsymbol{X} \cdot \boldsymbol{W}. \tag{23}$$

In GPF, the feature prompt $\hat{\boldsymbol{p}}$ is added to the feature vector for each node. Then the new node representation matrix with $\hat{\boldsymbol{p}}$ can be written as

$$\begin{aligned}
\boldsymbol{H}_{\hat{\boldsymbol{p}}} &= (\boldsymbol{A} + (1+\epsilon) \cdot \boldsymbol{I}) \cdot (\boldsymbol{X} + [1]^N \cdot \hat{\boldsymbol{p}}) \cdot \boldsymbol{W} \\
&= (\boldsymbol{A} + (1+\epsilon) \cdot \boldsymbol{I}) \cdot \boldsymbol{X} \cdot \boldsymbol{W} + (\boldsymbol{A} + (1+\epsilon) \cdot \boldsymbol{I}) \cdot [1]^N \cdot \hat{\boldsymbol{p}} \cdot \boldsymbol{W} \\
&= \boldsymbol{H} + (\boldsymbol{A} + (1+\epsilon) \cdot \boldsymbol{I}) \cdot [1]^N \cdot \hat{\boldsymbol{p}} \cdot \boldsymbol{W} \\
&= \boldsymbol{H} + [Deg_i + 1 + \epsilon]^N \cdot \hat{\boldsymbol{p}} \cdot \boldsymbol{W}
\end{aligned} \tag{24}$$

where $[1]^N \in \mathbb{R}^{N \times 1}$ represents an N-dimensional column vector with values of 1, $[Deg_i + 1 + \epsilon]^N \in \mathbb{R}^{N \times 1}$ represents an N-dimensional column vector with the value of $i$-th row is $Deg_i + 1 + \epsilon$, and $Deg_i$ represents the degree of node $v_i$.

In EdgePrompt, the prompt vector will be associated with each edge. Therefore, we can write the node representation matrix with edge prompt $\boldsymbol{p}$ by

$$\begin{aligned}
\boldsymbol{H}_{\boldsymbol{p}} &= \boldsymbol{A} \cdot (\boldsymbol{X} + [1]^N \cdot \boldsymbol{p}) \cdot \boldsymbol{W} + (1+\epsilon) \cdot \boldsymbol{X} \cdot \boldsymbol{W} \\
&= \boldsymbol{A} \cdot \boldsymbol{X} \cdot \boldsymbol{W} + \boldsymbol{A} \cdot [1]^N \cdot \boldsymbol{p} \cdot \boldsymbol{W} + (1+\epsilon) \cdot \boldsymbol{X} \cdot \boldsymbol{W} \\
&= \boldsymbol{H} + \boldsymbol{A} \cdot [1]^N \cdot \boldsymbol{p} \cdot \boldsymbol{W} \\
&= \boldsymbol{H} + [Deg_i]^N \cdot \boldsymbol{p} \cdot \boldsymbol{W}
\end{aligned} \tag{25}$$

To obtain the same graph representation, we have

$$\mathrm{Sum}(\boldsymbol{H}_{\hat{\boldsymbol{p}}}) = \mathrm{Sum}(\boldsymbol{H}_{\boldsymbol{p}}), \tag{26}$$

where $\mathrm{Sum}(\boldsymbol{H})$ computes the sum vector for each row vector of a matrix. We can simplify the above equation by

$$\begin{aligned}
&\mathrm{Sum}(\boldsymbol{H}_{\hat{\boldsymbol{p}}}) = \mathrm{Sum}(\boldsymbol{H}_{\boldsymbol{p}}) \\
\Rightarrow\ &\mathrm{Sum}(\boldsymbol{H} + [Deg_i + 1 + \epsilon]^N \cdot \hat{\boldsymbol{p}} \cdot \boldsymbol{W}) = \mathrm{Sum}(\boldsymbol{H} + [Deg_i]^N \cdot \boldsymbol{p} \cdot \boldsymbol{W}) \\
\Rightarrow\ &\mathrm{Sum}([Deg_i + 1 + \epsilon]^N \cdot \hat{\boldsymbol{p}} \cdot \boldsymbol{W}) = \mathrm{Sum}([Deg_i]^N \cdot \boldsymbol{p} \cdot \boldsymbol{W}) \\
\Rightarrow\ &(Deg + N + N \cdot \epsilon) \cdot \hat{\boldsymbol{p}} \cdot \boldsymbol{W} = Deg \cdot \boldsymbol{p} \cdot \boldsymbol{W}
\end{aligned} \tag{27}$$

where $Deg = \sum_{v_i \in gV} Deg_i$. To obtain the above equation, we only need

$$\boldsymbol{p} = \frac{Deg + N + N \cdot \epsilon}{Deg} \cdot \hat{\boldsymbol{p}}. \tag{28}$$

Therefore, for any feature prompt $\hat{\boldsymbol{p}}$, we can always find an edge prompt $\boldsymbol{p}$ in Equation (28) that satisfies Lemma 1.

**Extension to other GNN backbones.** Various GNN backbones can be expressed as $\boldsymbol{H} = \boldsymbol{S} \cdot \boldsymbol{X} \cdot \boldsymbol{W}$, where $\boldsymbol{S}$ is the diffusion matrix (Gasteiger et al., 2019). Different $\boldsymbol{S}$ only impact the coefficient before $\hat{\boldsymbol{p}}$ in Equation (28).

**Extension to multi-layer GNN models.** For multi-layer linear GNN models, the diffusion matrix $\boldsymbol{S}^{(l)}$ at each layer can be integrated as one overall $\boldsymbol{S}$. $\quad\square$

**Theorem 2.** *Given an input graph $\mathcal{G} = (\boldsymbol{X}, \boldsymbol{A})$ and its transformation $\mathcal{G}' = (\boldsymbol{X}', \boldsymbol{A}')$ by an arbitrary transformation function $\mathcal{T}$, there exists a set of edge prompt vectors $\{\boldsymbol{p}^{(1)}, \boldsymbol{p}^{(2)}, \cdots, \boldsymbol{p}^{(L)}\}$ in EdgePrompt that can satisfy*

$$f(\boldsymbol{X}, \boldsymbol{A}, \{\boldsymbol{p}^{(1)}, \cdots, \boldsymbol{p}^{(L)}\}) = f(\boldsymbol{X}', \boldsymbol{A}') \tag{29}$$

*for any pre-trained GNN model $f$.*

Table 4: Basic information and statistics of graph datasets adopted in our experiments.

| Dataset | #(Graphs) | #(Nodes) | #(Edges) | #(Features) | #(Classes) | Task Level |
|---|---|---|---|---|---|---|
| Cora | 1 | 2,708 | 10,556 | 1,433 | 7 | Node |
| CiteSeer | 1 | 3,327 | 9,104 | 3,703 | 6 | Node |
| Pubmed | 1 | 19,717 | 88,648 | 500 | 3 | Node |
| Flickr | 1 | 89,250 | 899,756 | 500 | 7 | Node |
| ogbn-arxiv | 1 | 169,343 | 1,166,243 | 128 | 40 | Node |

| Dataset | #(Graphs) | #(Avg. Nodes) | #(Avg. Edges) | #(Features) | #(Classes) | Task Level |
|---|---|---|---|---|---|---|
| ENZYMES | 600 | 32.63 | 124.27 | 3 | 6 | Graph |
| DD | 1,178 | 284.32 | 1,431.32 | 89 | 2 | Graph |
| NCI1 | 4,110 | 29.87 | 64.60 | 37 | 2 | Graph |
| NCI109 | 4,127 | 29.68 | 64.26 | 38 | 2 | Graph |
| Mutagenicity | 4,337 | 30.32 | 61.54 | 14 | 2 | Graph |

*Proof.* Given any feature prompts, Lemma 1 indicates that we can always find edge prompts that lead to the same representation of a graph for any pre-trained GNN models. Given Theorem 1 by (Fang et al., 2023), the input graph with a learnable feature prompt can always obtain the same representation as those of any transformed graphs. Therefore, we can conclude that our edge prompts in EdgePrompt have the capacity to obtain the representation equal to those of any transformed graphs for any pre-trained GNN models. □

## C  MORE DETAILS ABOUT EXPERIMENTAL SETUP

### C.1  DATASETS

Table 4 shows the basic information and statistics of graph datasets adopted in our experiments.

### C.2  PRE-TRAINING STRATEGIES

We provide more details about the four pre-training strategies adopted in our experiments.

- **GraphCL** (You et al., 2020) is a contrastive method for pre-training GNN models. The intuition of GraphCL is to maximize the agreement between two views of a graph perturbed by different data augmentations. We adopt node dropping and edge perturbation to generate two graph views. A GNN model generates two graph representations of the same graph. A nonlinear projection head will map the two graph representations to another latent space. The contrastive loss will be used to optimize the GNN model and the projection head.

- **SimGRACE** (Xia et al., 2022) is an augmentation-free contrastive method for GNN pre-training. We first construct a perturbed version of the GNN model by adding noise sampled from the Gaussian distribution. Given an input graph, the perturbed GNN model will generate its representation that forms a positive pair with that generated by the original GNN model.

- **EP-GPPT** (Sun et al., 2022a) pre-trains a GNN model using edge prediction. A set of edges in the original graph is randomly masked. The pre-training task is to predict whether a node pair is connected. Unconnected node pairs are randomly selected to form the negative samples in pre-training.

- **EP-GraphPrompt** (Liu et al., 2023) similarly uses edge prediction for GNn pre-training. Given a node in the input graph, we randomly sample one positive node from its neighbors and one negative node that does not link to it. The pre-training task is to maximize the similarity between the connected nodes while minimizing the similarity between the unconnected nodes.

Table 5: Average running time (seconds per epoch) on 5-shot node classification tasks over five datasets.

| Tuning Methods | Cora | CiteSeer | Pubmed | ogbn-arxiv | Flickr |
|---|---|---|---|---|---|
| Classifier Only | 0.116 | 0.136 | 0.663 | 1.186 | 5.156 |
| GPPT | 0.141 | 0.151 | 0.713 | 1.381 | 5.828 |
| GraphPrompt | 0.126 | 0.136 | 0.673 | 1.377 | 4.362 |
| All-in-one | 0.477 | 0.578 | 3.090 | 6.085 | 7.357 |
| GPF | 0.121 | 0.131 | 0.678 | 1.070 | 3.482 |
| GPF-plus | 0.116 | 0.131 | 0.668 | 1.075 | 3.427 |
| EdgePrompt | 0.121 | 0.136 | 0.693 | 1.106 | 3.824 |
| EdgePrompt+ | 0.146 | 0.156 | 0.804 | 1.377 | 5.894 |

Table 6: Average running time (seconds per epoch) on 50-shot graph classification tasks over five datasets.

| Tuning Methods | ENZYMES | DD | NCI1 | NCI109 | Mutagenicity |
|---|---|---|---|---|---|
| Classifier Only | 0.216 | 0.176 | 0.291 | 0.332 | 0.302 |
| GraphPrompt | 0.276 | 0.211 | 0.347 | 0.357 | 0.322 |
| All-in-one | 0.457 | 0.643 | 1.337 | 1.397 | 1.206 |
| GPF | 0.221 | 0.191 | 0.342 | 0.322 | 0.307 |
| GPF-plus | 0.231 | 0.191 | 0.347 | 0.296 | 0.312 |
| EdgePrompt | 0.226 | 0.196 | 0.347 | 0.296 | 0.317 |
| EdgePrompt+ | 0.332 | 0.302 | 0.442 | 0.382 | 0.402 |

# D MORE EXPERIMENTAL RESULTS

## D.1 RESULTS ON MODEL EFFICIENCY

Table 5 and Table 6 provide the average running time (seconds per epoch) for node classification and graph classification, respectively. From the two tables, we can observe that most graph prompt tuning method has similar computing time except All-in-one. All-in-one needs more time per epoch since it uses alternating strategies. EdgePrompt has almost the same efficiency as Classifier only without any prompts. In addition, EdgePrompt+ does not introduce significant computational cost.

## D.2 RESULTS ON GRAPH DATA WITH EDGE FEATURES

In our experiments, we conduct experiments over graph data without edge features. However, in the real world, many graphs may inherently have edge features. Our method is still compatible with this case. We report the performance of our method and other baselines over BACE and BBBP from the MoleculeNet dataset (Wu et al., 2018) in Table 7. From the table, we can observe that our method can outperform other baselines over the two datasets under two pre-training strategies.

## D.3 RESULTS WITH EDGE PROMPTS AT THE FIRST LAYER

Unlike previous studies, we learn prompt vectors at each layer of the pre-trained GNN model. This strategy can consistently avoid adverse information aggregated from different classes. For example, node $v_3$ in Figure 1 may receive adverse information from node $v_1$ when node $v_3$ and node $v_1$ are from different classes. If we learn edge prompts only at the first layer, node $v_3$ will still receive adverse information from node $v_1$ at the following layers. In contrast, our method in EdgePrompt+ instead learns layer-wise edge prompts, which can consistently avoid the above issue at each layer. We conduct experiments on our methods with edge prompts only at the first layer. Table 8 and Table 9 show the performance for node classification and graph classification, respectively. From the tables, we observe performance degradation in most cases, especially for EdgePrompt+. This observation validates our design of learning edge prompts at each layer of the pre-trained GNN model.

Table 7: Accuracy on 50-shot graph classification tasks over two datasets with edge features. The best-performing method is **bolded** and the runner-up underlined.

| Pre-training Strategies | Tuning Methods | BACE | BBBP |
|---|---|---|---|
| SimGRACE | Classifier Only | $57.62_{\pm 1.92}$ | $63.56_{\pm 1.03}$ |
| | GraphPrompt | $59.37_{\pm 0.53}$ | $63.39_{\pm 1.75}$ |
| | All-in-one | $56.73_{\pm 1.33}$ | $65.72_{\pm 3.48}$ |
| | GPF | $57.36_{\pm 1.52}$ | $63.89_{\pm 1.66}$ |
| | GPF-plus | $57.16_{\pm 2.21}$ | $64.17_{\pm 1.29}$ |
| | EdgePrompt | $58.12_{\pm 1.04}$ | $63.89_{\pm 1.26}$ |
| | EdgePrompt+ | $\mathbf{60.46_{\pm 2.63}}$ | $\mathbf{70.50_{\pm 1.92}}$ |
| EP-GraphPrompt | Classifier Only | $60.40_{\pm 1.03}$ | $66.17_{\pm 1.15}$ |
| | GraphPrompt | $61.69_{\pm 1.36}$ | $66.86_{\pm 0.70}$ |
| | All-in-one | $56.17_{\pm 1.54}$ | $61.72_{\pm 6.97}$ |
| | GPF | $60.89_{\pm 0.71}$ | $66.72_{\pm 0.84}$ |
| | GPF-plus | $61.39_{\pm 0.22}$ | $67.58_{\pm 0.67}$ |
| | EdgePrompt | $61.09_{\pm 1.22}$ | $66.94_{\pm 0.97}$ |
| | EdgePrompt+ | $\mathbf{64.66_{\pm 2.20}}$ | $\mathbf{72.75_{\pm 2.12}}$ |

Table 8: Accuracy on 5-shot node classification tasks over three datasets. The best-performing method is **bolded**.

| Pre-training Strategies | Tuning Methods | Cora | CiteSeer | Pubmed |
|---|---|---|---|---|
| GraphCL | EdgePrompt (first layer) | $57.74_{\pm 4.42}$ | $42.41_{\pm 3.21}$ | $67.33_{\pm 3.57}$ |
| | EdgePrompt | $58.60_{\pm 4.46}$ | $43.31_{\pm 3.23}$ | $\mathbf{67.76_{\pm 3.01}}$ |
| | EdgePrompt+ (first layer) | $61.66_{\pm 6.81}$ | $44.96_{\pm 2.63}$ | $67.54_{\pm 3.95}$ |
| | EdgePrompt+ | $\mathbf{62.88_{\pm 6.43}}$ | $\mathbf{46.20_{\pm 0.99}}$ | $67.41_{\pm 5.25}$ |
| EP-GPPT | EdgePrompt (first layer) | $36.74_{\pm 4.79}$ | $29.47_{\pm 3.16}$ | $47.98_{\pm 6.42}$ |
| | EdgePrompt | $37.26_{\pm 4.53}$ | $29.83_{\pm 1.01}$ | $47.20_{\pm 7.06}$ |
| | EdgePrompt+ (first layer) | $56.10_{\pm 6.39}$ | $42.10_{\pm 1.41}$ | $60.61_{\pm 7.57}$ |
| | EdgePrompt+ | $\mathbf{56.41_{\pm 3.62}}$ | $\mathbf{43.49_{\pm 2.62}}$ | $\mathbf{61.51_{\pm 4.91}}$ |

Table 9: Accuracy on 50-shot graph classification tasks over three datasets. The best-performing method is **bolded**.

| Pre-training Strategies | Tuning Methods | ENZYMES | NCI1 | NCI109 |
|---|---|---|---|---|
| SimGRACE | EdgePrompt (first layer) | $28.83_{\pm 1.74}$ | $61.58_{\pm 2.71}$ | $61.82_{\pm 1.15}$ |
| | EdgePrompt | $29.33_{\pm 2.30}$ | $62.02_{\pm 3.02}$ | $62.02_{\pm 1.03}$ |
| | EdgePrompt+ (first layer) | $28.58_{\pm 2.45}$ | $61.81_{\pm 3.03}$ | $62.36_{\pm 0.98}$ |
| | EdgePrompt+ | $\mathbf{32.67_{\pm 2.53}}$ | $\mathbf{67.07_{\pm 1.96}}$ | $\mathbf{66.53_{\pm 1.30}}$ |
| EP-GraphPrompt | EdgePrompt (first layer) | $30.75_{\pm 1.03}$ | $61.81_{\pm 2.57}$ | $62.07_{\pm 1.42}$ |
| | EdgePrompt | $30.80_{\pm 2.09}$ | $61.75_{\pm 2.49}$ | $62.33_{\pm 1.65}$ |
| | EdgePrompt+ (first layer) | $31.92_{\pm 1.41}$ | $62.07_{\pm 2.64}$ | $61.66_{\pm 1.64}$ |
| | EdgePrompt+ | $\mathbf{33.27_{\pm 2.71}}$ | $\mathbf{65.06_{\pm 1.84}}$ | $\mathbf{64.64_{\pm 1.57}}$ |

## D.4 MORE RESULTS ON CONVERGENCE PERFORMANCE

Figure 5 illustrates the accuracy curves of our method and the baselines under two pre-training strategies for graph classification.

## D.5 RESULTS WITH DIFFERENT SHOTS

We conduct experiments with different shots. Table 10 shows the performance for 10-shot node classification tasks. In addition, we also conduct experiments for 100-shot graph classification tasks

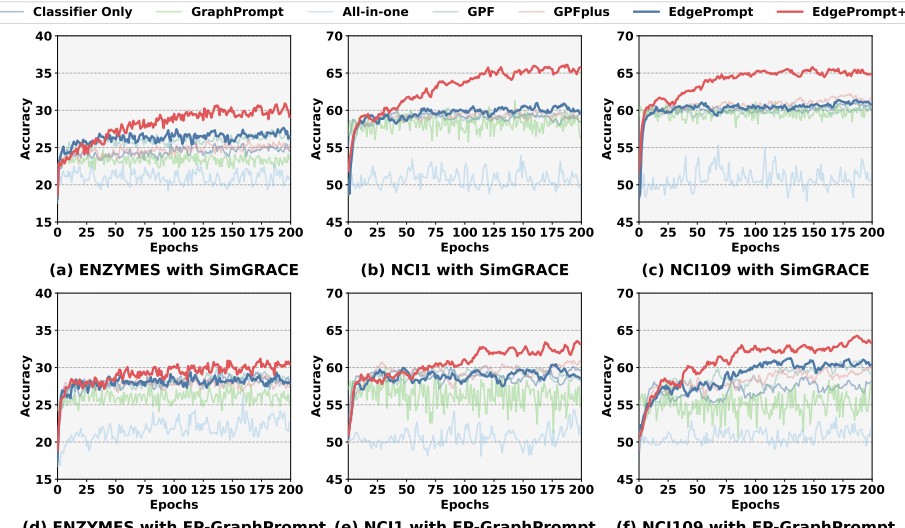

Figure 5: Convergence speeds of different methods.

and report the results in Table 11. Since ENZYMES uses up all graphs as the training samples in the 100-shot setting, we run experiments on the remaining four datasets.

# E   FUTURE WORKS

In the future, we will investigate the performance of our method under more pre-training strategies, such as DGI (Veličković et al., 2019), InfoGraph (Sun et al., 2020), GraphMAE (Hou et al., 2022). In addition, we will explore other designs for edge prompts, such as conditional prompting Yu et al. (2024c); Zhou et al. (2022). Furthermore, we will also study how to adapt our method for heterogeneous graphs.

Table 10: Accuracy on 10-shot node classification tasks over five datasets. The best-performing method is **bolded** and the runner-up is underlined.

| Pre-training Strategies | Tuning Methods | Cora | CiteSeer | Pubmed | ogbn-arxiv | Flickr |
|---|---|---|---|---|---|---|
| GraphCL | Classifier Only | $65.43_{\pm2.29}$ | $43.97_{\pm4.39}$ | $68.23_{\pm1.05}$ | $26.78_{\pm1.66}$ | $30.34_{\pm2.33}$ |
| | GPPT | $58.38_{\pm3.43}$ | $44.65_{\pm8.47}$ | $67.71_{\pm6.35}$ | $26.54_{\pm3.69}$ | $28.80_{\pm5.93}$ |
| | GraphPrompt | $63.55_{\pm2.49}$ | $46.17_{\pm3.07}$ | $67.73_{\pm1.83}$ | $25.51_{\pm1.00}$ | $26.74_{\pm1.90}$ |
| | ALL-in-one | $51.57_{\pm7.11}$ | $43.31_{\pm2.78}$ | $61.20_{\pm2.83}$ | $21.84_{\pm2.45}$ | $24.63_{\pm3.75}$ |
| | GPF | $70.06_{\pm1.88}$ | $47.34_{\pm4.22}$ | $70.70_{\pm1.30}$ | $27.43_{\pm1.51}$ | $27.59_{\pm2.21}$ |
| | GPF-plus | $65.32_{\pm1.93}$ | $43.97_{\pm3.97}$ | $\underline{68.32}_{\pm0.91}$ | $26.75_{\pm1.32}$ | $29.81_{\pm1.43}$ |
| | EdgePrompt | $\underline{70.20}_{\pm1.77}$ | $\underline{47.85}_{\pm4.19}$ | $70.54_{\pm1.55}$ | $27.52_{\pm1.20}$ | $28.58_{\pm2.50}$ |
| | EdgePrompt+ | $\mathbf{74.27}_{\pm3.46}$ | $\mathbf{52.93}_{\pm4.20}$ | $\mathbf{72.70}_{\pm2.50}$ | $\mathbf{28.79}_{\pm1.21}$ | $\mathbf{30.74}_{\pm2.30}$ |
| SimGRACE | Classifier Only | $62.18_{\pm3.15}$ | $45.62_{\pm3.74}$ | $60.60_{\pm1.87}$ | $27.09_{\pm0.93}$ | $30.35_{\pm1.90}$ |
| | GPPT | $60.00_{\pm5.11}$ | $40.27_{\pm7.11}$ | $62.16_{\pm6.35}$ | $27.26_{\pm3.44}$ | $30.31_{\pm6.39}$ |
| | GraphPrompt | $59.26_{\pm2.06}$ | $47.22_{\pm3.37}$ | $62.53_{\pm1.71}$ | $25.66_{\pm0.83}$ | $30.16_{\pm1.22}$ |
| | ALL-in-one | $49.83_{\pm2.90}$ | $43.94_{\pm2.83}$ | $59.99_{\pm1.99}$ | $20.03_{\pm3.03}$ | $29.64_{\pm3.72}$ |
| | GPF | $67.73_{\pm4.06}$ | $49.08_{\pm3.36}$ | $63.58_{\pm1.65}$ | $\underline{27.92}_{\pm0.94}$ | $32.96_{\pm3.94}$ |
| | GPF-plus | $62.22_{\pm3.36}$ | $45.44_{\pm4.15}$ | $60.67_{\pm1.77}$ | $27.09_{\pm0.82}$ | $\mathbf{33.89}_{\pm3.31}$ |
| | EdgePrompt | $\underline{68.28}_{\pm4.05}$ | $\underline{49.29}_{\pm3.45}$ | $63.67_{\pm1.66}$ | $27.88_{\pm1.00}$ | $\underline{33.56}_{\pm3.58}$ |
| | EdgePrompt+ | $\mathbf{72.57}_{\pm3.50}$ | $\mathbf{52.78}_{\pm3.29}$ | $\mathbf{69.56}_{\pm2.58}$ | $\mathbf{28.70}_{\pm0.91}$ | $32.17_{\pm2.77}$ |
| EP-GPPT | Classifier Only | $34.12_{\pm3.25}$ | $28.42_{\pm3.32}$ | $45.05_{\pm4.12}$ | $15.94_{\pm1.80}$ | $31.96_{\pm5.48}$ |
| | GPPT | $48.43_{\pm6.16}$ | $35.94_{\pm6.09}$ | $56.50_{\pm9.44}$ | $\mathbf{23.58}_{\pm1.84}$ | $29.58_{\pm6.81}$ |
| | GraphPrompt | $35.08_{\pm1.43}$ | $28.12_{\pm1.56}$ | $48.71_{\pm5.28}$ | $13.38_{\pm1.84}$ | $29.08_{\pm3.51}$ |
| | ALL-in-one | $35.12_{\pm1.62}$ | $27.19_{\pm2.63}$ | $47.11_{\pm1.56}$ | $16.57_{\pm0.37}$ | $\mathbf{32.30}_{\pm2.42}$ |
| | GPF | $49.61_{\pm0.40}$ | $35.19_{\pm2.46}$ | $50.52_{\pm2.75}$ | $22.48_{\pm2.21}$ | $31.60_{\pm5.54}$ |
| | GPF-plus | $33.60_{\pm2.34}$ | $28.18_{\pm3.31}$ | $45.13_{\pm4.67}$ | $16.07_{\pm1.82}$ | $30.81_{\pm7.60}$ |
| | EdgePrompt | $\underline{50.43}_{\pm0.83}$ | $\underline{34.56}_{\pm3.04}$ | $\underline{50.90}_{\pm2.51}$ | $\underline{22.61}_{\pm2.21}$ | $30.80_{\pm6.58}$ |
| | EdgePrompt+ | $\mathbf{69.65}_{\pm6.44}$ | $\mathbf{50.74}_{\pm2.80}$ | $\mathbf{60.83}_{\pm4.36}$ | $21.66_{\pm2.06}$ | $30.78_{\pm5.75}$ |
| EP-GraphPrompt | Classifier Only | $68.17_{\pm3.25}$ | $47.94_{\pm3.58}$ | $75.49_{\pm1.79}$ | $36.69_{\pm0.80}$ | $31.38_{\pm8.08}$ |
| | GPPT | $68.93_{\pm4.55}$ | $48.83_{\pm8.45}$ | $74.78_{\pm6.81}$ | $25.65_{\pm3.55}$ | $32.85_{\pm3.09}$ |
| | GraphPrompt | $68.95_{\pm2.57}$ | $50.26_{\pm2.21}$ | $75.73_{\pm1.40}$ | $36.86_{\pm0.84}$ | $\underline{30.39}_{\pm5.31}$ |
| | ALL-in-one | $57.74_{\pm3.19}$ | $46.14_{\pm5.72}$ | $74.24_{\pm3.04}$ | $22.84_{\pm2.60}$ | $30.61_{\pm5.28}$ |
| | GPF | $\underline{72.24}_{\pm2.92}$ | $51.07_{\pm3.76}$ | $\mathbf{77.77}_{\pm2.42}$ | $36.91_{\pm1.09}$ | $29.74_{\pm8.94}$ |
| | GPF-plus | $68.32_{\pm3.75}$ | $48.33_{\pm3.62}$ | $75.57_{\pm1.73}$ | $36.63_{\pm1.08}$ | $29.40_{\pm8.30}$ |
| | EdgePrompt | $72.20_{\pm2.47}$ | $\underline{51.40}_{\pm3.60}$ | $\underline{77.35}_{\pm2.52}$ | $\underline{37.16}_{\pm1.18}$ | $32.01_{\pm4.61}$ |
| | EdgePrompt+ | $\mathbf{75.08}_{\pm3.11}$ | $\mathbf{56.09}_{\pm2.63}$ | $76.66_{\pm2.07}$ | $\mathbf{37.28}_{\pm1.43}$ | $\mathbf{34.49}_{\pm7.10}$ |

Table 11: Accuracy on 100-shot graph classification tasks over four datasets. The best-performing method is **bolded** and the runner-up underlined.

| Pre-training Strategies | Tuning Methods | DD | NCI1 | NCI109 | Mutagenicity |
|---|---|---|---|---|---|
| GraphCL | Classifier Only | $63.23_{\pm 1.42}$ | $62.03_{\pm 1.60}$ | $62.18_{\pm 1.59}$ | $68.29_{\pm 1.26}$ |
| | GraphPrompt | $62.80_{\pm 1.15}$ | $62.17_{\pm 1.21}$ | $61.79_{\pm 0.99}$ | $68.14_{\pm 0.94}$ |
| | All-in-one | $66.33_{\pm 1.78}$ | $60.69_{\pm 1.15}$ | $62.00_{\pm 0.37}$ | $64.39_{\pm 2.74}$ |
| | GPF | $66.75_{\pm 1.14}$ | $62.48_{\pm 1.65}$ | $61.98_{\pm 0.97}$ | $68.41_{\pm 1.60}$ |
| | GPF-plus | $\mathbf{68.49_{\pm 1.98}}$ | $\underline{65.39_{\pm 2.27}}$ | $\underline{64.85_{\pm 1.41}}$ | $68.78_{\pm 1.22}$ |
| | EdgePrompt | $66.96_{\pm 1.05}$ | $63.84_{\pm 1.75}$ | $62.42_{\pm 0.91}$ | $68.69_{\pm 1.59}$ |
| | EdgePrompt+ | $\underline{67.81_{\pm 1.49}}$ | $\mathbf{67.54_{\pm 1.40}}$ | $\mathbf{67.94_{\pm 0.81}}$ | $\mathbf{70.52_{\pm 0.58}}$ |
| SimGRACE | Classifier Only | $63.74_{\pm 0.96}$ | $63.27_{\pm 1.68}$ | $63.20_{\pm 2.00}$ | $67.65_{\pm 1.28}$ |
| | GraphPrompt | $63.82_{\pm 0.95}$ | $63.58_{\pm 1.35}$ | $61.52_{\pm 1.10}$ | $67.97_{\pm 0.97}$ |
| | All-in-one | $\mathbf{68.92_{\pm 0.61}}$ | $59.94_{\pm 2.12}$ | $62.79_{\pm 0.48}$ | $64.47_{\pm 2.02}$ |
| | GPF | $65.90_{\pm 2.02}$ | $64.32_{\pm 1.55}$ | $63.48_{\pm 1.82}$ | $67.44_{\pm 1.01}$ |
| | GPF-plus | $67.04_{\pm 1.53}$ | $\underline{65.28_{\pm 2.05}}$ | $\underline{64.72_{\pm 1.64}}$ | $67.95_{\pm 0.88}$ |
| | EdgePrompt | $65.99_{\pm 2.29}$ | $65.09_{\pm 1.46}$ | $63.65_{\pm 1.69}$ | $68.23_{\pm 0.81}$ |
| | EdgePrompt+ | $\underline{68.03_{\pm 1.85}}$ | $\mathbf{67.24_{\pm 1.87}}$ | $\mathbf{67.59_{\pm 1.63}}$ | $\mathbf{69.50_{\pm 0.54}}$ |
| EP-GPPT | Classifier Only | $62.68_{\pm 1.93}$ | $58.47_{\pm 1.07}$ | $63.24_{\pm 0.67}$ | $66.57_{\pm 1.26}$ |
| | GraphPrompt | $60.55_{\pm 1.53}$ | $59.11_{\pm 0.66}$ | $62.76_{\pm 0.85}$ | $67.12_{\pm 1.42}$ |
| | All-in-one | $62.51_{\pm 1.25}$ | $59.06_{\pm 1.47}$ | $62.07_{\pm 0.96}$ | $65.04_{\pm 0.84}$ |
| | GPF | $63.82_{\pm 3.44}$ | $59.31_{\pm 1.49}$ | $63.75_{\pm 0.63}$ | $66.64_{\pm 1.34}$ |
| | GPF-plus | $\mathbf{68.87_{\pm 2.80}}$ | $\underline{64.48_{\pm 2.57}}$ | $\underline{65.10_{\pm 0.81}}$ | $69.00_{\pm 1.10}$ |
| | EdgePrompt | $64.84_{\pm 3.27}$ | $60.57_{\pm 1.57}$ | $63.60_{\pm 0.67}$ | $67.15_{\pm 1.40}$ |
| | EdgePrompt+ | $\underline{68.28_{\pm 2.03}}$ | $\mathbf{66.28_{\pm 1.15}}$ | $\mathbf{66.72_{\pm 1.34}}$ | $\mathbf{71.52_{\pm 1.58}}$ |
| EP-GraphPrompt | Classifier Only | $65.95_{\pm 1.79}$ | $62.88_{\pm 0.81}$ | $62.02_{\pm 2.27}$ | $67.39_{\pm 0.80}$ |
| | GraphPrompt | $66.24_{\pm 1.70}$ | $62.93_{\pm 0.97}$ | $62.27_{\pm 1.05}$ | $67.67_{\pm 0.74}$ |
| | All-in-one | $66.45_{\pm 1.24}$ | $60.73_{\pm 1.46}$ | $58.56_{\pm 0.70}$ | $66.53_{\pm 1.10}$ |
| | GPF | $68.37_{\pm 2.66}$ | $62.68_{\pm 1.45}$ | $63.75_{\pm 1.67}$ | $67.98_{\pm 0.97}$ |
| | GPF-plus | $\underline{68.89_{\pm 3.93}}$ | $\underline{63.91_{\pm 0.99}}$ | $63.55_{\pm 2.42}$ | $67.84_{\pm 0.96}$ |
| | EdgePrompt | $67.81_{\pm 3.64}$ | $63.33_{\pm 1.40}$ | $\underline{64.00_{\pm 1.91}}$ | $68.04_{\pm 1.07}$ |
| | EdgePrompt+ | $\mathbf{69.04_{\pm 2.96}}$ | $\mathbf{66.80_{\pm 0.55}}$ | $\mathbf{65.94_{\pm 1.15}}$ | $\mathbf{71.48_{\pm 1.89}}$ |

