# OpenReview forum: "Edge Prompt Tuning for Graph Neural Networks"
_ICLR.cc/2025/Conference — ICLR 2025 Poster_

### Official Review · Reviewer_t54u · 2024-10-29

**Soundness:** 2
**Presentation:** 3
**Contribution:** 2
**Rating:** 6
**Confidence:** 4

**Summary:**

The paper presents EdgePrompt, a method that enhances pre-trained GNNs for downstream tasks by using learnable prompt vectors on edges. EdgePrompt+ further customizes these vectors for individual edges. This approach improves graph structural representation and is compatible with various GNN architectures. Experiments on multiple datasets show its effectiveness over existing methods for node and graph classification tasks.

**Strengths:**

1. The paper is well-organized, with clear points, and is easy to follow.
2. The effectiveness of EdgePrompt is theoretically guaranteed, and it performs excellently in downstream tasks.

**Weaknesses:**

1. The motivation for constructing EdgePrompt is insufficient. Why is it necessary to design EdgePrompt under graph prompt tuning? What core problem does EdgePrompt address compared to existing graph prompt tuning methods? What are its advantages?
2. Compared to ALL-in-one and GPF, EdgePrompt and EdgePrompt+ set different prompt vectors $p^{(l)}$ for each layer. What are the benefits of this design? Both All-in-one and GPF only add prompt vectors in the first layer to reduce dependency on the specific structure of the model. EdgePrompt lacks such advantages, and the paper does not explore the reasoning behind this design. Furthermore, the experimental section does not include relevant comparisons to demonstrate the necessity of setting different prompt vectors for each layer.
3. The datasets included in the experimental section do not contain initial edge features, which raises doubts about the effectiveness of EdgePrompt on graphs that inherently have edge features. If the original graph already contains edge features, how should EdgePrompt be integrated with these edge features? What would its performance be like in that case?
4. The downstream tasks involved in the experiments are limited to node classification and graph classification, with other graph tasks such as link prediction and node regression not being included.

**Questions:**

Please refer to the points I mentioned in the weakness part.

---

> ### Author Response · Authors · 2024-11-19
> **Author Response to Reviewer t54u (1/2)**
>
> We deeply appreciate your insightful comments to make our paper better. We hope that we can address all your concerns in our point-by-point responses.
>
> ---
> - W1: The motivation for constructing EdgePrompt is insufficient. Why is it necessary to design EdgePrompt under graph prompt tuning? What core problem does EdgePrompt address compared to existing graph prompt tuning methods? What are its advantages?
>
> - **R1**: As indicated in Introduction, the existing graph prompt tuning methods mainly focus on learning graph prompts at the node level and overlook the crucial role of edges in graph prompt design. As a result, they cannot effectively enhance pre-trained GNN models in capturing complex graph structural information for downstream tasks. In fact, graph structures are the essence of graph data that differentiates graph data from image or text data. Motivated by this, we propose EdgePrompt and its advanced version EdgePrompt+ in this study. Our method innovatively learns graph prompts at the edge level and explicitly captures graph structural information to enhance pre-trained GNN models for downstream tasks.
>
> ---
> - W2: Compared to ALL-in-one and GPF, EdgePrompt and EdgePrompt+ set different prompt vectors $p^{(l)}$ for each layer. What are the benefits of this design? Both All-in-one and GPF only add prompt vectors in the first layer to reduce dependency on the specific structure of the model. EdgePrompt lacks such advantages, and the paper does not explore the reasoning behind this design. Furthermore, the experimental section does not include relevant comparisons to demonstrate the necessity of setting different prompt vectors for each layer.
>
> - **R2**: Thanks for pointing it out. The intuition of edge prompts for each layer can be illustrated in Figure 1. Node $v_3$ may receive adverse information from node $v_1$ when node $v_3$ and node $v_1$ are from different classes. If we learn edge prompts only in the first layer, node $v_3$ will still receive adverse information from node $v_1$ in the following layers. In contrast, our method instead learns layer-wise edge prompts, which can consistently avoid the above issue in each layer. We provide the results of EdgePrompt and EdgePrompt+ with edge prompts only in the first layer in the following tables. We can observe performance degradation in some cases compared with their original versions with edge prompts in each layer, especially for EdgePrompt+. In addition, we would like to note that learning layer-wise prompts has been adopted by recent studies [1, 2] from other areas. If layer-wise prompts are not allowed, learning prompts in the first layer can be an alternative approach. We have added the discussion in our paper (See Appendix D.3 in our revised PDF).
>
>   &nbsp;
>   [1] Visual Prompt Tuning. *ECCV* 2022.
>   &nbsp;
>   [2] MaPLe: Multi-Modal Prompt Learning. *CVPR* 2023.
>
>   |Pre-training: GraphCL     |  Cora      |   CiteSeer  |   PubMed   |
>   |----------------------------|--------------|--------------|--------------|
>   |  EdgePrompt (first layer)  |57.74$\pm$4.42|42.41$\pm$3.21|67.33$\pm$3.57|
>   |       EdgePrompt      |58.60$\pm$4.46|43.31$\pm$3.23|**67.76$\pm$3.01**|
>   |  EdgePrompt+ (first layer) |61.66$\pm$6.81|44.96$\pm$2.63|67.54$\pm$3.95|
>   |       EdgePrompt+     |**62.88$\pm$6.43**|**46.20$\pm$0.99**|67.41$\pm$5.25|
>
>   |Pre-training: EP-GPPT     |  Cora      |   CiteSeer  |   PubMed   |
>   |----------------------------|--------------|--------------|--------------|
>   |  EdgePrompt (first layer)  |36.74$\pm$4.79|29.47$\pm$3.16|47.98$\pm$6.42|
>   |       EdgePrompt      |37.26$\pm$4.53|29.83$\pm$1.01|47.20$\pm$7.06|
>   |  EdgePrompt+ (first layer) |56.10$\pm$6.39|42.10$\pm$1.41|60.61$\pm$7.57|
>   |       EdgePrompt+     |**56.41$\pm$3.62**|**43.49$\pm$2.62**|**61.51$\pm$4.91**|
>
>   | Pre-training: SimGRACE    |  ENZYMES    |    NCI1    |   NCI109   |
>   |----------------------------|--------------|--------------|--------------|
>   |  EdgePrompt (first layer)  |28.83$\pm$1.74|61.58$\pm$2.71|61.82$\pm$1.15|
>   |       EdgePrompt      |29.33$\pm$2.30|62.02$\pm$3.02|62.02$\pm$1.03|
>   |  EdgePrompt+ (first layer) |28.58$\pm$2.45|61.81$\pm$3.03|62.36$\pm$0.98|
>   |       EdgePrompt+     |**32.67$\pm$2.53**|**67.07$\pm$1.96**|**66.53$\pm$1.30**|
>
>   |Pre-training: EP-GraphPrompt|  ENZYMES    |    NCI1    |   NCI109   |
>   |----------------------------|--------------|--------------|--------------|
>   |  EdgePrompt (first layer)  |30.75$\pm$1.03|61.81$\pm$2.57|62.07$\pm$1.42|
>   |       EdgePrompt      |30.80$\pm$2.09|61.75$\pm$2.49|62.33$\pm$1.65|
>   |  EdgePrompt+ (first layer) |31.92$\pm$1.41|62.07$\pm$2.64|61.66$\pm$1.64|
>   |       EdgePrompt+     |**33.27$\pm$2.71**|**65.06$\pm$1.84**|**64.64$\pm$1.57**|

---

> ### Author Response · Authors · 2024-11-19
> **Author Response to Reviewer t54u (2/2)**
>
> ---
> - W3: The datasets included in the experimental section do not contain initial edge features, which raises doubts about the effectiveness of EdgePrompt on graphs that inherently have edge features. If the original graph already contains edge features, how should EdgePrompt be integrated with these edge features? What would its performance be like in that case?
>
> - **R3**: Thanks for bringing this up. When handling graph data with edge features, we can still use the current strategy in EdgePrompt and EdgePrompt+ to learn prompts vectors. The only difference is that edge features/embeddings will be aggregated along with prompt vectors. To evaluate the performance of our method over graph data with edge features, we conduct experiments over BACE and BBBP from the MoleculeNet dataset [1]. The following two tables report the accuracy of our method and other baselines. We have added the discussion in our paper (see Appendix D.2 in our revised PDF).
>
>   &nbsp;
>   [1] MoleculeNet: a benchmark for molecular machine learning. *Chemical science* 2018.
>
>   | Pre-training: SimGRACE|  BACE  |   BBBP    |
>   |-----------------------|-----------|-----------|
>   | Classifier Only     |  57.62$\pm$1.92  | 63.56$\pm$1.03 |
>   | GraphPrompt        |  59.37$\pm$0.53  | 63.39$\pm$1.75 |
>   | All-in-one        | 56.73$\pm$1.33   | 65.72$\pm$3.48 |
>   | GPF             | 57.36$\pm$1.52  | 63.89$\pm$1.66 |
>   | GPF-plus          | 57.16$\pm$2.21  | 64.17$\pm$1.29 |
>   |  EdgePrompt        | 58.12$\pm$1.04  | 63.89$\pm$1.26 |
>   | EdgePrompt+        | **60.46$\pm$2.63**  | **70.50$\pm$1.92** |
>
>   | Pre-training: EP-GraphPrompt|  BACE  |   BBBP    |
>   |-----------------------|-----------|-----------|
>   | Classifier Only     |  60.40$\pm$1.03  | 66.17$\pm$1.15 |
>   | GraphPrompt        |  61.69$\pm$1.36  | 66.86$\pm$0.70 |
>   | All-in-one        | 56.17$\pm$1.54  | 61.72$\pm$6.97 |
>   | GPF             | 60.89$\pm$0.71  | 66.72$\pm$0.84 |
>   | GPF-plus          | 61.39$\pm$0.22  | 67.58$\pm$0.67 |
>   |  EdgePrompt        | 61.09$\pm$1.22  | 66.94$\pm$0.97 |
>   | EdgePrompt+        | **64.66$\pm$2.20**  | **72.75$\pm$2.12** |
>
> ---
> - W4: The downstream tasks involved in the experiments are limited to node classification and graph classification, with other graph tasks such as link prediction and node regression not being included.
>
> - **R4**: Thanks for bringing this up. We would like to clarify that **we provide results for node classification and graph classification as downstream tasks, as these tasks are commonly used in previous studies on graph pre-training and graph prompt tuning.** For example, in graph pre-training studies, GraphCL, SimGRACE, and InfoGraph use graph classification, while DGI uses node classification. As for graph prompt tuning studies, GPPT focuses on node classification, GPF focuses on graph classification, and GraphPrompt focuses both. Therefore, we follow these studies to conduct experiments in our study. In addtion, we would like to argue that link prediction as the downstream task may be incompatible with the "pre-training, adaptation" scheme. During pre-training, GNN models should be trained via self-supervised learning. If the downstream task is link prediction, however, we will directly have label information (i.e., whether an edge exists between a node pair) in graph data. In this case, we can simply follow an end-to-end manner by using link prediction to train GNN models and then making inferences (we guess it is the reason why the above studies choose not to include the results of link prediction as the downstream task). Considering this, we believe node classification and graph classification are proper and sufficient for performance evaluation in our experiments.

---

> ### Comment · Reviewer_t54u · 2024-11-23
>
> Thank you for the author's patient responses. I have thoroughly read all the author's replies as well as the feedback from other reviewers. The additional experiments have made the paper more convincing. I will maintain the score I have given.

---

> > ### Author Response · Authors · 2024-11-23
> > **Thanks for your feedback**
> >
> > We thank you for your valuable suggestions and feedback. Your evaluation is very important to us. If you think you still have any other unsolved concerns, we will be more than happy to provide more clarifications.
> >
> > Thanks,
> > Authors of Submission 4905

---

### Official Review · Reviewer_7GLy · 2024-10-30

**Soundness:** 3
**Presentation:** 4
**Contribution:** 3
**Rating:** 6
**Confidence:** 4

**Summary:**

The paper proposes EdgePrompt, a graph prompt tuning method that enhances GNNs by learning prompt vectors for edges, improving graph representations. EdgePrompt integrates these edge prompts through message passing, outperforming existing methods across ten datasets under four pre-training strategies.

**Strengths:**

1. The paper is well-motivated. It's important to integrate structural knowledge in prompt learning.
2. The authors conducted extensive experiments, demonstrating the effectiveness of the proposed methods.
3. The authors provide theoretical analysis, further proving the effectiveness of the proposed methods.
4. The paper is well written and easy to follow.

**Weaknesses:**

1. **Inaccurate statement**: GraphPrompt [1] is not based on a specific pre-training strategy. As shown in GraphPrompt+ [2], all contrastive learning pre-training methods can be unified as subgraph similarity calculations. The link prediction used in [1] can be replaced by other methods.
2. **Missing related work**: GraphPrompt+ [1] also adds prompt vectors to each layer of the pre-trained graph encoder, which should be discussed and compared.
3. **Unclear explanation of anchor prompts in EdgePrompt+**: It is unclear what the anchor prompts in EdgePrompt+ represent. In my opinion, anchor prompts are introduced to address the overfitting problem caused by directly learning edge-specific prompts for different edges, but there lacks a explanation for the meaning of the anchor prompts. A more reasonable and effective solution could be conditional prompting [3,4], which I highly recommend the authors explore in future work.


[1] Liu et al. "Graphprompt: Unifying pre-training and downstream tasks for graph neural networks." Proceedings of the ACM Web Conference 2023. 2023.\
[2] Yu et al. "Generalized graph prompt: Toward a unification of pre-training and downstream tasks on graphs." IEEE Transactions on Knowledge and Data Engineering (2024).\
[3] Zhou et al. "Conditional prompt learning for vision-language models." Proceedings of the IEEE/CVF conference on computer vision and pattern recognition. 2022.\
[4] Yu  et al. "Non-Homophilic Graph Pre-Training and Prompt Learning." arXiv preprint arXiv:2408.12594 (2024).

**Questions:**

See weaknesses.

---

> ### Author Response · Authors · 2024-11-19
> **Author Response to Reviewer 7GLy**
>
> We deeply appreciate your insightful comments to make our paper better. We hope that we can address all your concerns in our point-by-point responses.
>
> ---
> - W1: Inaccurate statement: GraphPrompt [1] is not based on a specific pre-training strategy. As shown in GraphPrompt+ [2], all contrastive learning pre-training methods can be unified as subgraph similarity calculations. The link prediction used in [1] can be replaced by other methods.
>
> - **R1**: Thanks for pointing it out. In fact, we indeed had a tough time identifying the compatibility of GraphPrompt with different pre-training strategies. While we notice that the authors of GraphPrompt use link prediction for pre-training as a component of GraphPrompt, the adaptation phase in GraphPrompt does not explicitly require any specific information from the pre-training phase. We conjecture that the authors may think using link prediction is the most suitable pre-training task regarding the loss function for prompt tuning in GraphPrompt. We thank the reviewer for bringing up its variant GraphPrompt+, a great complement to the compatibility of GraphPrompt with different pre-training strategies. We have modified Table 1 to correct the inaccurate statement about GraphPrompt in our revised PDF.
>
> ---
> - W2: Missing related work: GraphPrompt+ [1] also adds prompt vectors to each layer of the pre-trained graph encoder, which should be discussed and compared.
>
> - **R2**: Thanks for bringing this up. We have added GraphPrompt+ in Table 1 of our revised PDF.
>
> ---
> - W3: Unclear explanation of anchor prompts in EdgePrompt+: It is unclear what the anchor prompts in EdgePrompt+ represent. In my opinion, anchor prompts are introduced to address the overfitting problem caused by directly learning edge-specific prompts for different edges, but there lacks a explanation for the meaning of the anchor prompts. A more reasonable and effective solution could be conditional prompting [3,4], which I highly recommend the authors explore in future work.
>
> - **R3**: Thanks for bringing this up. We agree that anchor prompts can address the overfitting problem caused by learning an independent prompt for each edge. However, we also want to emphasize that learning independent edge-specific prompts encounters critical supervision starvation for node classification, especially in the few-shot setting. As explained in Section 4.2, if one edge is not involved in computing the representations of any labeled nodes, its edge prompt will not be updated at all. In this case, we cannot learn anything on this edge prompt. To overcome this issue, we propose to learn the prompt vectors as a weighted average of multiple anchor prompts. We may regard these anchors prompts as a set of basis prompts shared by all edges. Therefore, an edge prompt is a combination of these basis prompts in the prompt space. In this case, each edge just needs to learn the weight scores by Equation (5) and (6). We appreciate your suggestion using conditional prompting. We have mentioned it as our future work in our paper (see Appendix E in our revised PDF).

---

> > ### Comment · Reviewer_7GLy · 2024-11-29
> >
> > Thank you for your responses, which have addressed my concerns. As reflected in my score, I hold a positive attitude toward this paper.

---

> > > ### Author Response · Authors · 2024-11-29
> > >
> > > Dear Reviewer 7GLy,
> > >
> > > Thanks for your reply. We are glad that our repsonses have adressed your concerns. We appreciate your positive attitude toward our paper. If you have any other questions or suggestions to improve our paper, we are always willing to provide more explanations.
> > >
> > > Best,
> > > Authors of Submission 4905

---

> ### Author Response · Authors · 2024-11-25
> **Looking Forward to Your Feedback**
>
> Dear Reviewer 7GLy,
>
> Thank you again for reviewing our paper. Your evaluation is very important to our paper. According to your valuable comments, we have modified Table 1 (about GraphPrompt and GraphPrompt+) and Future Works (about conditional prompting) in our PDF. We believe that these modifications and our clarifications have addressed all your concerns — in light of this, **we hope you could consider raising your rating score**. If you have any further questions, we are willing to provide more explanations.
>
> Thanks,
> Authors of Submission 4905

---

### Official Review · Reviewer_WRHJ · 2024-11-02

**Soundness:** 2
**Presentation:** 2
**Contribution:** 2
**Rating:** 5
**Confidence:** 4

**Summary:**

This paper introduces EdgePrompt, a new graph prompt tuning method that improves graph representation for downstream tasks by learning edge-specific prompts, enhancing the performance of pre-trained GNNs. Extensive experiments show EdgePrompt’s effectiveness across various datasets and pre-training strategies, outperforming several baseline methods.

**Strengths:**

1. EdgePrompt improves the adaptation of pre-trained GNN models for downstream tasks by introducing edge-level prompts, which helps bridge the objective gap between pre-training and downstream tasks..
2. Extensive experiments on multiple datasets and pre-training strategies demonstrate the method’s effectiveness, showing better performance compared to existing graph prompt tuning approaches.

**Weaknesses:**

1. EdgePrompt uses shared prompt vectors, which may not capture the different relationships between edges well. This can limit the model’s ability to use all the information in the graph.
2. EdgePrompt+ adds multiple anchor prompts and score calculations, which can make the model more complex. This can lead to higher computational costs, making it harder to use in larger graphs.
3. The method struggles with few-shot learning because most edges lack supervision. This can reduce the model’s performance in real-world tasks where labeled data is limited.

**Questions:**

How can the performance of EdgePrompt be improved in scenarios with limited labeled data to enhance its effectiveness in node classification tasks?

---

> ### Author Response · Authors · 2024-11-18
> **Author Response to Reviewer WRHJ**
>
> We sincerely appreciate your efforts to review our paper and provide insightful suggestions. We hope our following point-by-point clarifications can address your concerns.
>
> ---
> - W1: EdgePrompt uses shared prompt vectors, which may not capture the different relationships between edges well. This can limit the model’s ability to use all the information in the graph.
>
>
> - **R1**: Yes. That is why we propose an advanced version of EdgePrompt, e.g., EdgePrompt+, in this study to overcome this issue.
>
> ---
> - W2: EdgePrompt+ adds multiple anchor prompts and score calculations, which can make the model more complex. This can lead to higher computational costs, making it harder to use in larger graphs.
>
> - **R2**: Thanks for bringing this up. We would like to argue that **our method does not introduce significant computational cost**. Our method only needs $L$-hop local graphs to compute edge prompts, which is **scalable in larger graphs** like ogbn-arxiv. Here, we provide the results of running time (seconds per epoch) for each method in the following two tables. We have added the discussion in our paper (See Appendix D.1 in our revised PDF).
>
>
>   | Tuning Methods |  Cora    | CiteSeer  | Pubmed   |ogbn-arxiv | Flickr   |
>   |----------------|-----------|-----------|-----------|-----------|-----------|
>   | Classifier Only|  0.116   |  0.136   |  0.663   |  1.186   |  5.156   |
>   | GPPT        |  0.141   |  0.151   |  0.713   |  1.381   |  5.828   |
>   | GraphPrompt   |  0.126   |  0.136   |  0.673   |  1.377   |  4.362   |
>   | All-in-one    |  0.477   |  0.578   |  3.090   |  6.085   |  7.357   |
>   |  GPF        |  0.121   |  0.131   |  0.678   |  1.070   |  3.482   |
>   | GPF-plus     |  0.116   |  0.131   |  0.668   |  1.075   |  3.427   |
>   |  EdgePrompt   |  0.121   |  0.136   |  0.693   |  1.106   |  3.824   |
>   | EdgePrompt+   |  0.146   |  0.156   |  0.804   |  1.377   |  5.894   |
>
>   | Tuning Methods |  ENZYMES  |   DD    |  NCI1    | NCI109   |Mutagenicity|
>   |----------------|-----------|-----------|-----------|-----------|-----------|
>   | Classifier Only|  0.216   |  0.176   |  0.291   |  0.332   |  0.302   |
>   | GraphPrompt   |  0.276   |  0.211   |  0.347   |  0.357   |  0.322   |
>   | All-in-one    |  0.457   |  0.643   |  1.337   |  1.397   |  1.206   |
>   |  GPF        |  0.221   |  0.191   |  0.342   |  0.322   |  0.307   |
>   | GPF-plus     |  0.231   |  0.191   |  0.347   |  0.296   |  0.312   |
>   |  EdgePrompt   |  0.226   |  0.196   |  0.347   |  0.296   |  0.317   |
>   | EdgePrompt+   |  0.332   |  0.302   |  0.442   |  0.382   |  0.402   |
>
>
> ---
> - W3: The method struggles with few-shot learning because most edges lack supervision. This can reduce the model’s performance in real-world tasks where labeled data is limited.
>
> - **R3**: We would like to clarify that our method aims to deal with the few-shot setting. Therefore, **our method does not struggle with few-shot learning and does not encounter performance degradation when labeled data is limited**. The lack of supervision is exactly the motivation of our design in EdgePrompt+ to handle the few-shot learning. In addition, our experiments are based on the few-shot setting. Experimental results demonstrate that our method outperforms other baselines under the few-shot setting.
>
> ---
> - Q1: How can the performance of EdgePrompt be improved in scenarios with limited labeled data to enhance its effectiveness in node classification tasks?
>
> - **A1**: EdgePrompt can be enhanced by its advanced version - EdgePrompt+. Our theoretical analysis and empirical results validate the superiority of EdgePrompt+.

---

> > ### Comment · Reviewer_WRHJ · 2024-11-24
> >
> > Thank you for your detailed and thoughtful response! While I appreciate the clarifications provided, I still believe the incremental novelty of this paper remains marginal. The method proposed appears to be a natural extension of the GPF [1] approach. For example, GPF introduced the use of a shared vector as a node feature prompt, and to enhance its performance, GPF-plus introduced the concept of a basic vector. I’ve quoted the relevant part of the original paper below for reference:
> >
> > > Similarly to GPF, the prompted features $X^*$ replace the initial features $X$ and are processed by the pre-trained model. However, such a design is not universally suitable for all scenarios. For instance, when training graphs have different scales (i.e., varying node numbers), it is challenging to train such a series of $p_i$. Additionally, when dealing with large-scale input graphs, this design requires a substantial amount of storage resources due to its $O(N)$ learnable parameters. To address these issues, we introduce an attention mechanism in the generation of $p_i$, making GPF-plus more parameter-efficient and capable of handling graphs with different scales. In practice, we train only $k$ independent basis vectors $p_b$, where $k$ is a hyper-parameter that can be adjusted based on the downstream dataset. To obtain $p_i$ for node $v_i$, we utilize attentive aggregation of these basis vectors with the assistance of $k$ learnable linear projections.
> >
> > In conclusion, I do not believe this paper meets the high standards expected for ICLR, as the contributions appear to build incrementally on prior work without introducing sufficiently novel elements.
> >
> > [1] Fang, Taoran, et al. "Universal prompt tuning for graph neural networks." Advances in Neural Information Processing Systems 36 (2024).

---

> > > ### Author Response · Authors · 2024-11-30
> > > **Looking Forward to Your Feedback**
> > >
> > > Dear Reviewer WRHJ,
> > >
> > > Thank you again for reviewing our paper. Your evaluation is very important to our paper.
> > >
> > > We have provided more clarification about the novelty and contributions of our work in the previous comment. We hope it can fully address your concern. We are willing to provide more explanations if you have any further questions.
> > >
> > > Best,
> > > Authors of Submission 4905

---

> ### Author Response · Authors · 2024-11-24
>
> Thanks for your feedback. We would like to clarify that the novelty of our work lies in the following two aspects.
>
> - **A novel graph prompting method from a fundamentally different perspective of edges**. As illustrated in Table 1, GPF and GPF-plus design graph prompts on node features $X$. However, such a desgin does not capture the uniqueness of graph data, as they do not integrate any structure information in graph prompts. In other words, their design can be seamlessly used on Euclidean data, such as images. In contrast, our method finds a new direction by designing graph prompts from the perspective of edges $\mathcal{E}$, which has never been investigated by previous studies. As we know, it is graph structures that differentiate graph data from image data. As indicated in line 271, GPF-plus can be regarded as a special case of our method. Therefore, we believe that **only our study handles the key issue of graph data on edges when designing graph prompts**.
>
> - **Theoretical analysis on node-level tasks**. In our study, we provide theoretical analysis on the effectiveness of our study for node-level tasks. Our analysis indicates that our design can effectively enhance the pre-trained GNN model for node classification, while GPF and GPF-plus fail to achieve this.
> To the best of our knowledge, **our paper is the first work to provide theoretical analysis of graph prompt tuning methods on node-level tasks.** The theoretical analysis is also taken as an important contribution by other reviewers (Reviewer 7GLy, t54u).
>
> Considering this, we would like to argue that our study is novel and make a great contribution to the community in terms of prompt design and theoretical analysis. We believe this study will attract and inspire following studies in graph prompt tuning to explore how to design graph prompts at the edge level in the future.
>
> We hope the above clarification solves your new concern. We are willing to provide more explanations if you have any further questions.

---

> ### Author Response · Authors · 2024-12-01
> **A kind reminder**
>
> Dear Reviewer WRHJ,
>
> Thank you again for reviewing our paper. As the discussion period is ending in two days, we are eager to know whether our following clarifications have adressed your concerns. We hope these clarifications can still be considered for your evaluation, which is very important to us. We are willing to provide more explanations if you have any further questions.
>
> Thanks,
> Authors of Submission 4905

---

> ### Author Response · Authors · 2024-12-02
> **We are still waiting for your reply**
>
> Dear Reviewer WRHJ,
>
> Thank you again for reviewing our paper. As the discussion period is ending in less than 24 hours, we are eager to know whether our following clarifications have adressed your concerns. We hope these clarifications can still be considered for your evaluation, which is very important to us. We are willing to provide more explanations if you have any further questions.
>
> Thanks,
> Authors of Submission 4905

---

### Official Review · Reviewer_D2XH · 2024-11-04

**Soundness:** 2
**Presentation:** 3
**Contribution:** 2
**Rating:** 5
**Confidence:** 3

**Summary:**

Recent graph prompt tuning methods have proven effective in adapting pre-trained GNNs to downstream tasks. However, they often overlook the crucial role of edges in graph prompt design. To address this research gap, this submission introduces a new graph prompt tuning method focused on edges, called EdgePrompt. Nevertheless, despite emphasizing the importance of edges in graphs, the authors make an overly strong assumption by considering only a single type of edge. Additionally, the paper does not address edge-related tasks, which significantly undermines the overall contribution and impact of the work.

**Strengths:**

S1. Clear motivation and presentation.

S2. The proposed method can be integrated with existing pre-trained GNNs.

**Weaknesses:**

**Weakness**

W1. The unclear statements regarding the edge-level aspect weaken the paper’s contributions.

W2. The authors need to further elaborate on the technical contributions.

W3. More experiments are needed to better support the superiority of the proposed method.

**Concerns**

C1. As a study focused on edge-level prompt tuning, the assumption that there is only one type of edge could significantly undermine the contributions and claims of this paper. In line 154, the modeling of the adjacency matrix, $\mathbf{A} \in \{0,1\}^{N \times N}$, implies that the paper does not target multi-relational graphs. However, compared to other node-level graph prompting systems, the proposed edge-level graph prompting method could be more suitable for graphs with multiple edge types. The authors may need to clarify this in the submission.

C2. Since this work emphasizes edge-level prompt tuning, it would be beneficial for the authors to explore edge-related tasks, such as edge classification and link prediction, to further expand the scope of the paper.

C2-1. In many real-world scenarios, studying edge-level tasks is highly relevant because the space of edge types can evolve over time. For example, in a social network, a newly introduced user interaction feature might require predicting new edge types using a trained GNN.

C2-2. If the research on edge-level tasks is beyond the scope of current pre-trained GNNs (i.e., no existing pre-trained GNNs focus on edge-level tasks), the authors should clarify this limitation in the submission.

C3. The core Equation (4) in EdgePrompt+ appears overly similar to existing work, which may diminish the paper’s technical contribution. In CompGCN [1], the operation of weighting relation embeddings based on relation base embeddings has already been shown to be simple and parameter-efficient. Therefore, the authors should elaborate on the unique technical contributions of their method.

Minor Concerns:

C4. More classic and promising pre-trained GNNs, such as Infomax, EdgePred, AttrMasking, MGSSL, GraphMAE, and Mole-BERT, could be included in the experimental section. At the very least, the authors should discuss these models and explain why they are excluded from comparison.

C5. Figure 2 presents convergence speeds in terms of the number of epochs. The authors should also analyze the efficiency of the proposed method using learning curves or running time comparisons.


**Reference**

[1] COMPOSITION-BASED MULTI-RELATIONAL GRAPH CONVOLUTIONAL NETWORKS, ICLR 2020.

**Questions:**

Please focus on answering concerns C1-C3.

---

> ### Author Response · Authors · 2024-11-18
> **Author Response to Reviewer D2XH (1/3)**
>
> We sincerely appreciate your efforts to review our paper and provide valuable suggestions.
>
> ---
> - Before our point-by-point clarifications, we first would like to provide an overall summary of this study in plain words to avoid misunderstanding. Graph prompt tuning methods aim to learn "something" extra (i.e., prompt vectors) to adapt pre-trained GNN models for downstream tasks while keeping the pre-trained GNN models frozen. For example, GPF learns "something" extra on node features, and GraphPrompt learns "something" extra on node representations. Under the same setting, we hope to answer the question: can we adapt pre-trained GNN models by learning "something" extra on edges? As we know, it is graph structures that differentiate graph data from image data or text data. In this study, we propose EdgePrompt and its advanced version EdgePrompt+ that learn prompt vectors on edges. EdgePrompt learns shared prompt vectors for all the edges, while EdgePrompt+ learns customized, unique prompt vectors for each edge. In a nutshell, this study targets the same goal under the same settings as previous studies (like GPF and GraphPrompt) but designs a different strategy from a novel perspective of edges.
>
> ---
> - C1. As a study focused on edge-level prompt tuning, the assumption that there is only one type of edge could significantly undermine the contributions and claims of this paper. In line 154, the modeling of the adjacency matrix, $\mathbf{A} \in \{0, 1\}^{N \times N}$, implies that the paper does not target multi-relational graphs. However, compared to other node-level graph prompting systems, the proposed edge-level graph prompting method could be more suitable for graphs with multiple edge types. The authors may need to clarify this in the submission.
>
> - **R1**: Thanks for bringing this up. We would like to emphasize that **our study shares the same assumption with previous graph prompt tuning studies**, such as GPF, GraphPrompt, and All-in-one. In addition, **our graph prompt tuning method on edge prompts is irrelevant with edge types**. Instead, we aim to adapt the pre-trained GNN models by learning customized prompt vectors for each edge. As indicated in Section 4.2, the prompt vectors are edge-specific, which means every edge will have its unique prompt vectors. Therefore, we will have $|\mathcal{E}|$ different $e^{(l)}$ for a graph with $|\mathcal{E}|$ edges. We believe it is completely different from multi-relational graphs where we may hope to model edge types.

---

> > ### Comment · Reviewer_D2XH · 2024-11-29
> > **ACK**
> >
> > Thanks for the authors’ response and clarifications. I understand the authors’ intention in this work, which is to address a task similar to those in previous related studies but from the perspective of edges. This explains why the authors consistently emphasized the alignment of their assumptions with those in prior works in their responses to my concerns. However, this raises an important point of discussion: if the assumptions are indeed aligned, the edge-focused approach presented in this work may frequently lead to similar concerns. The authors may overemphasize the role of edges in node and graph-level tasks while overlooking the intrinsic nature of edge-related tasks. It would be beneficial for the authors to address this issue carefully when framing their paper. Additionally, merely emphasizing the time cost per epoch seems to create curiosity about the overall running time required.

---

> > > ### Author Response · Authors · 2024-12-01
> > > **Looking Forward to Your Feedback**
> > >
> > > Dear Reviewer D2XH,
> > >
> > > Thank you again for reviewing our paper. As the discussion period is ending soon, we are eager to know whether our following clarifications have adressed your concerns. We hope these clarifications can still be considered for your evaluation, which is very important to us. We are willing to provide more explanations if you have any further questions.
> > >
> > > Thanks,
> > > Authors of Submission 4905

---

> > > ### Author Response · Authors · 2024-12-02
> > > **The discussion period is ending soon**
> > >
> > > Dear Reviewer D2XH,
> > >
> > > Thank you again for reviewing our paper. As the discussion period is ending in 24 hours, we are eager to learn whether our answers have addressed your concerns. We are looking forward to your feedback and happy to answer any extra questions.
> > >
> > > Best,
> > > Authors of Submission 4905

---

> ### Author Response · Authors · 2024-11-18
> **Author Response to Reviewer D2XH (2/3)**
>
> ---
> - C2. Since this work emphasizes edge-level prompt tuning, it would be beneficial for the authors to explore edge-related tasks, such as edge classification and link prediction, to further expand the scope of the paper.
>
> - **R2**: Thanks for bringing this up. We would like to clarify that designing edge-level graph prompts does not mean we particularly focus on edge-level tasks. Instead, our design aims to enhance pre-trained GNN models in capturing graph structural information for diverse downstream tasks. **We provide results for node classification and graph classification as downstream tasks, as we are following previous studies on graph pre-training and graph prompt tuning.** For example, in graph pre-training studies, GraphCL, SimGRACE, and InfoGraph use graph classification, while DGI uses node classification. As for graph prompt tuning studies, GPPT focuses on node classification, GPF focuses on graph classification, and GraphPrompt focuses both. We follow these studies to conduct experiments in our study.
>
> ---
> - C2-1. In many real-world scenarios, studying edge-level tasks is highly relevant because the space of edge types can evolve over time. For example, in a social network, a newly introduced user interaction feature might require predicting new edge types using a trained GNN.
>
> - **R2-1**: Thanks for bringing this up. We would like to clarify again that **this study is irrelevant with edge types**.
>
> ---
> - C2-2. If the research on edge-level tasks is beyond the scope of current pre-trained GNNs (i.e., no existing pre-trained GNNs focus on edge-level tasks), the authors should clarify this limitation in the submission.
>
> - **R2-2**: Thanks for bringing this up. Current pre-trained GNNs mainly focus on node classification and graph classification. **Even if we regard it as a limitation, it is about graph pre-training studies but not about the graph prompt tuning stage.** In addtion, we would like to argue that **link prediction as the downstream task may be incompatible with the "pre-training, adaptation" scheme**. During pre-training, GNN models should be trained via self-supervised learning. If the downstream task is link prediction, however, we will directly have label information (i.e., whether an edge exists between a node pair) in graph data. In this case, we can simply follow an end-to-end manner by using link prediction to train GNN models and then making inferences (we guess it is the reason why previous studies choose not to include the results of link prediction as the downstream task). Considering this, we believe node classification and graph classification are proper and sufficient for performance evaluation in our experiments.
>
> ---
> - C3. The core Equation (4) in EdgePrompt+ appears overly similar to existing work, which may diminish the paper’s technical contribution. In CompGCN [1], the operation of weighting relation embeddings based on relation base embeddings has already been shown to be simple and parameter-efficient. Therefore, the authors should elaborate on the unique technical contributions of their method.
>
> - **R3**: Thanks for bringing this up. We would like to clarify that **they are different in two aspects**. First, they basically have different targets. Equation (4) in EdgePrompt+ computes edge-specific prompt vectors, while CompGCN aims to compute relation-specific embeddings. Second, they obtain weights in different ways. The score vector in Equation (4) is obtained through a score function, while CompGCN takes weights as independent variables.

---

> ### Author Response · Authors · 2024-11-18
> **Author Response to Reviewer D2XH (3/3)**
>
> ---
> - C4. More classic and promising pre-trained GNNs, such as Infomax, EdgePred, AttrMasking, MGSSL, GraphMAE, and Mole-BERT, could be included in the experimental section. At the very least, the authors should discuss these models and explain why they are excluded from comparison.
>
> - **R4**: Thanks for bringing this up. We would to clarify that our framework is compatible with various pre-training methods. As summarized in Related Work, the existing numerous pre-training methods can be roughly categorized into two genres: contrastive methods and generative methods. For example, EdgePred, AttrMasking, and GraphMAE are generative methods while Infomax is a contrastive one. In our experiments, we select two contrastive methods (GraphCL and SimGRACE) and two generative methods (EP-GPPT and EP-GraphPrompt). **We adopt them because they are representative pre-training methods in the two genres and are also used by other graph prompt tuning studies.** For example, All-in-one uses GraphCL and SimGRACE. In addition, EP-GPPT and EP-GraphPrompt are edge prediction-based pre-training methods proposed by GPPT and GraphPrompt, respectively. Therefore, we believe the adopted four pre-training methods are inclusive and fair for performance comparison of different graph prompt tuning methods. We will explore the performance of our framework under other pre-training methods in the future (see Appendix E in our revised PDF).
>
> ---
> - C5. Figure 2 presents convergence speeds in terms of the number of epochs. The authors should also analyze the efficiency of the proposed method using learning curves or running time comparisons.
>
> - **R5**: Thanks for bringing this up. We would like to emphasize that most deep learning papers (e.g., GPPT and GPF in our baselines) report performance per epoch since the evaluation is conducted after each epoch. Therefore, we follow the common evaluation scheme in our experiment. In addition, we provide the results of running time (seconds per epoch) for each method in the following two tables. From the tables, we can observe that our method does not introduce significant computational cost. We have added the discussion in our paper (see Appendix D.1 in our revised PDF).
>
>
>   | Tuning Methods |  Cora    | CiteSeer  | Pubmed   |ogbn-arxiv | Flickr   |
>   |----------------|-----------|-----------|-----------|-----------|-----------|
>   | Classifier Only|  0.116   |  0.136   |  0.663   |  1.186   |  5.156   |
>   | GPPT        |  0.141   |  0.151   |  0.713   |  1.381   |  5.828   |
>   | GraphPrompt   |  0.126   |  0.136   |  0.673   |  1.377   |  4.362   |
>   | All-in-one    |  0.477   |  0.578   |  3.090   |  6.085   |  7.357   |
>   |  GPF        |  0.121   |  0.131   |  0.678   |  1.070   |  3.482   |
>   | GPF-plus     |  0.116   |  0.131   |  0.668   |  1.075   |  3.427   |
>   |  EdgePrompt   |  0.121   |  0.136   |  0.693   |  1.106   |  3.824   |
>   | EdgePrompt+   |  0.146   |  0.156   |  0.804   |  1.377   |  5.894   |
>
>   | Tuning Methods |  ENZYMES  |   DD    |  NCI1    | NCI109   |Mutagenicity|
>   |----------------|-----------|-----------|-----------|-----------|-----------|
>   | Classifier Only|  0.216   |  0.176   |  0.291   |  0.332   |  0.302   |
>   | GraphPrompt   |  0.276   |  0.211   |  0.347   |  0.357   |  0.322   |
>   | All-in-one    |  0.457   |  0.643   |  1.337   |  1.397   |  1.206   |
>   |  GPF        |  0.221   |  0.191   |  0.342   |  0.322   |  0.307   |
>   | GPF-plus     |  0.231   |  0.191   |  0.347   |  0.296   |  0.312   |
>   |  EdgePrompt   |  0.226   |  0.196   |  0.347   |  0.296   |  0.317   |
>   | EdgePrompt+   |  0.332   |  0.302   |  0.442   |  0.382   |  0.402   |

---

> ### Author Response · Authors · 2024-11-25
> **Looking Forward to Your Feedback**
>
> Dear Reviewer D2XH,
>
> Thank you again for reviewing our paper. Your evaluation is very important to our paper. We believe that our point-by-point clarifications have addressed all your concerns — in light of this, **we hope you could consider raising your rating score**. If you have any further questions, we are willing to provide more explanations.
>
> Thanks,
> Authors of Submission 4905

---

> ### Author Response · Authors · 2024-11-29
> **A kind reminder**
>
> Dear Reviewer D2XH,
>
> Thank you again for reviewing our paper. As the discussion phase is ending in three days, we are eager to learn whether our answers have addressed your concerns. We are looking forward to your feedback and happy to answer any extra questions.
>
> Best,
> Authors of Submission 4905

---

> ### Author Response · Authors · 2024-11-29
>
> Dear Reviewer D2XH,
>
> Thanks for your reply.
>
> Our study follows the same assumptions and focuses on the same downstream tasks (i.e., node classification and graph classification) in previous studies. **The only difference is how we design graph prompts (i.e., edge-based prompts in our study vs node-based prompts in previous studies).**
>
> We would like to argue that the importance of graph prompts on edges has been discussed in our theoretical analysis Section 4.3 and 4.4. **We believe that graph prompts on edges are significant for node-level and graph-level tasks and should not be underestimated.** We use Figure 1 to illustrate why edge-based approach is better than node-based approaches. Our edge-based method enables neighboring nodes to receive different finer learned prompt vectors from one node, which cannot be achieved by node-based method.
>
> As for time cost, we provide the results of time cost per epoch. Since we run each experiment 200 epochs, **the overall running time is 200$\times$(seconds per epoch) for every method.**
>
> We hope the above clarification can better address your concerns. We will be happy to answer any further questions you may have.

---

> ### Author Response · Authors · 2024-11-30
>
> In addition, we conduct experiments on edge classification for each method. Edge labels are constructed following All-in-one. The following tables report the accuracy of these methods under GraphCL and EP-GraphPrompt. According to the tables, we can observe that our method can still outperform other baselines for edge classification in most cases.
>
> We hope the new results can still be considered for your evaluation. We will add them in our revised version.
>
>   | Pre-training: GraphCL |  Cora  |   CiteSeer    | Pubmed   |
>   |-----------------------|-----------|-----------|-----------|
>   | Classifier Only     | 32.77$\pm$0.78     | 27.56$\pm$1.38     | 40.48$\pm$2.31 |
>   | GraphPrompt        | 35.79$\pm$1.85     | 31.87$\pm$1.91     | 45.39$\pm$1.22 |
>   | All-in-one        | 34.85$\pm$1.89      | 28.67$\pm$1.29     | 43.26$\pm$1.50 |
>   | GPF             | 36.88$\pm$1.53     | 29.32$\pm$1.88     | 46.76$\pm$1.47 |
>   | GPF-plus          | 40.34$\pm$1.82     | 32.55$\pm$3.13     | 47.53$\pm$2.13 |
>   |  EdgePrompt        | 36.78$\pm$1.54     | 29.18$\pm$1.91     | 45.98$\pm$2.70 |
>   | EdgePrompt+        | **41.95$\pm$2.35**     | **33.86$\pm$2.95**     | **47.89$\pm$3.01** |
>
>   | Pre-training: EP-GraphPrompt |  Cora  |   CiteSeer    | Pubmed   |
>   |-----------------------|-----------|-----------|-----------|
>   | Classifier Only     | 39.40$\pm$1.87     | 33.05$\pm$1.30     | 52.45$\pm$3.73 |
>   | GraphPrompt        | 42.86$\pm$2.52     | 34.89$\pm$1.98     | 52.96$\pm$3.19 |
>   | All-in-one        |  40.68$\pm$1.29     | 33.77$\pm$3.68     | 51.08$\pm$2.99 |
>   | GPF             | 41.24$\pm$2.72     | 33.27$\pm$2.30     | 52.61$\pm$2.67 |
>   | GPF-plus          | 43.18$\pm$2.61     | 34.79$\pm$2.78     | **55.05$\pm$3.06** |
>   |  EdgePrompt        | 41.12$\pm$2.56     | 33.24$\pm$2.20     | 49.18$\pm$2.63 |
>   | EdgePrompt+        | **43.93$\pm$2.00**     | **35.20$\pm$2.63**     | 53.19$\pm$3.73 |

---

### Author Response · Authors · 2024-11-20
**Summary of Rebuttal Revision**

We sincerely thank all the reviewers for their efforts to review our work. In response to the valuable feedback, we have made several major updates to our manuscript, as outlined below:
1. We have corrected the pre-training compatibility of GraphPrompt and added GraphPrompt+ for comparison (see the $\color{blue}{\text{blue}}$ part in Table 1);
2. We have added new results on model efficiency, results on graph data with edge features, and results with edge prompts at the first layer (see the $\color{red}{\text{red}}$ part in Appendix D);
3. We have added future works about experiments under more pre-training strategies, other designs for edge prompt (e.g., conditional prompting), adaptation for heterogeneous graphs (see the $\color{purple}{\text{purple}}$ part in Appendix E).

We hope that the revised manuscript can help address the concerns and resolve the issues raised by the reviewers.

Best,
Authors of Submission 4905

---

### Author Response · Authors · 2024-11-23
**General Response to All Reviewers**

Dear reviewers,

We sincerely appreciate your time and effort to review our paper.
We are happy to see the reviewers' recognition of our paper's strengths, including ***clear motivation*** (Reviewer D2XH, 7GLy), ***theoretical analysis*** (Reviewer 7GLy, t54u), ***comprehensive experimental evaluations*** (Reviewer WRHJ, 7GLy, t54u), and ***good presentation*** (Reviewer D2XH, 7GLy, t54u).

You insightful suggestions are important to our paper. We have provided point-by-point responses to reviewers' comments and updated corresponding sections in our PDF. We think our responses have fully addressed your concerns  — in light of this, **we hope you consider raising your score**. Please let us know in case there are any other concerns, and if so, we would be happy to respond.

Best,
Authors of Submission 4905

---

### Meta-Review · Area_Chair_RvXa · 2024-12-17

**Metareview:**

The paper proposes EdgePrompt, a graph prompt tuning method that enhances GNNs by learning prompt vectors for edges, improving graph representations. The reviewers agree that is well-organized, with clear points, and is easy to follow. The effectiveness of EdgePrompt is theoretically guaranteed, and it performs excellently in downstream tasks. Although, some of the reviewers noted that since this work emphasizes edge-level prompt tuning, it would be beneficial for the authors to explore edge-related tasks, such as edge classification and link prediction, to further expand the scope of the paper.

**Additional Comments On Reviewer Discussion:**

Reviewer WRHJ noted that the method proposed appears to be a natural extension of the GPF [1] approach. For example, GPF introduced the use of a shared vector as a node feature prompt, and to enhance its performance, GPF-plus introduced the concept of a basic vector.
The authors replied that GPF and GPF-plus design graph prompts on node features，and they believe that only their study handles the key issue of graph data on edges when designing graph prompts.
[1] Fang, Taoran, et al. "Universal prompt tuning for graph neural networks." Advances in Neural Information Processing Systems 36 (2024).

---

### Decision · Program_Chairs · 2025-01-22

Accept (Poster)